# LLM-GUIDED SPATIO-TEMPORAL DISEASE PROGRESSION MODELLING

## ABSTRACT

Understanding the interactions between biomarkers across brain regions during disease progression is essential for unravelling the mechanisms underlying neurodegenerative disease. For example, in Alzheimer's Disease (AD) and other neurodegenerative conditions, there are typically two kinds of methods to contract disease trajectory. Existing mechanistic models describe how variables interact with each other spatiotemporally within a dynamical system driven by an underlying biological substrate often based on brain connectivity. However, such methods typically grossly oversimplify the complex relationship between brain connectivity and brain pathology appearance and propagation. Meanwhile, pure data-driven approaches for inferring these relationships from time series face challenges with convergence, identifiability, and interpretability. We present a novel framework that bridges this gap by using Large Language Models (LLMs) as expert guides to learn disease progression from irregular longitudinal patient data. Our method simultaneously optimizes two components: 1) estimating the temporal positioning of patient data along a common disease trajectory, and 2) discovering the graph structure that captures spatiotemporal relationships between brain regions. By leveraging multiple LLMs as domain experts, our approach achieves faster convergence, improved stability, and better interpretability compared to existing methods. When applied to modelling tau-pathology propagation in the brain, our framework demonstrates superior prediction accuracy while revealing additional disease-driving factors beyond traditional connectivity measures. This work represents the first application of LLM-guided graph learning for modelling neurodegenerative disease progression in the brain from cross-sectional and short longitudinal imaging data.

**keywords** LLM, spatio-temporal modelling, disease progression

## 1 INTRODUCTION

Neurodegenerative diseases exhibit a progressive propagation of pathology throughout the brain Busche & Hyman (2020). Understanding the long-term progression of these diseases from their early to advanced stages is a key challenge for developing disease-modifying treatments. However, constraints of real-world patient data acquisition often hamper such efforts. Since medical scans can be expensive or pose potential health risks, data is often collected irregularly and over a narrow time frame. Accordingly, a set of modern computational approaches, known collectively as data-driven disease progression models Fonteijn et al. (2012); Young et al. (2014), has emerged to address the challenge of estimating population-level trajectories of change from such sparse and irregularly sampled patient data sets.

Mechanistic disease progression models Zhou et al. (2012); Raj et al. (2012b); Seguin et al. (2023b); Garbarino et al. (2019); Young et al. (2024b) simulate disease evolution using hypothetical mechanisms from patient data. For neurodegenerative diseases like Alzheimer's, these models capture spatiotemporal dynamics through two components: i) a graph that approximates the ability of each region's pathology occupancy to cause pathology appearance in each other region and ii) a mechanism of propagation between regions given that set of graph links. Network diffusion models (NDMs) Raj et al. (2012b); Weickenmeier et al. (2018) represent a key class of these models, assuming pathology spreads by diffusing along structural brain connections from MRI. While current approaches use brain connectivity measures as proxies for graph link strength, this oversimplifies the complex relationship between disease pathophysiology and brain connectivity, which can be measured differently

and changes during disease progression. Recent approaches, e.g. Garbarino et al. (2019); He et al. (2023); Thompson et al. (2024) acknowledge this limitation and aim to combine NDMs with multiple underlying propagation mechanisms including structural/ functional connectivity and/or proximity. To date, however, researchers have explored only simple linear combinations that are unlikely to capture the intricate interplay between these factors.

Unlike mechanistic modelling, where the equations of the models are explicit and the embedded graph is predefined, data-driven graph learning for time series methods aims to infer relationships among multiple variables, often represented using a graph. This has potential applications in mechanistic disease progression modelling by estimating the graph that drives pathology propagation in a more data-driven way. Related usage of structure learning methods includes Bellot et al. Bellot et al. (2021), who introduce a score-based learning algorithm using penalized Neural Ordinary Differential Equations (ODEs) to infer variable dependent relationships from irregularly-sampled, multivariate longitudinal data. However, fully data-driven graph inference faces several challenges. Identifiability remains a significant hurdle. Additionally, stable and rapid convergence of the estimated graph becomes increasingly difficult for high-dimensional data. Furthermore, data-driven methods often generate graphs that lack interpretability.

To address these limitations from both the mechanistic and data-driven graph learning sides, we consider using Large Language Models (LLMs) as expert guides to enforce the graph inference with expert knowledge Kıcıman et al. (2023); Abdulaal et al. (2023). Specifically, we develop a novel disease progression model for neurodegenerative diseases, which simultaneously a) uncovers the interactions between regional markers of brain pathology and b) reconstructs the temporal trajectory of those markers from irregularly sampled spatio-temporal data. We use a mixture of LLMs as experts for graph inference from time series, bringing these emerging ideas into this context for the first time. However, current data-driven graph learning methods for longitudinal data, designed for scenarios with known timestamps, fall short when applied to disease progression modelling in neurodegenerative diseases where the temporal position of each data point is unknown a-priori, as the timeline is learned during model estimation. Thus, our approach uniquely tackles the challenge of simultaneously optimizing the placement of each data point along the disease progression timeline while simultaneously inferring the inner relationships that inform the trajectory. In summary, we propose a novel framework guided by a mixture of LLMs as experts to model the interactions of biomarkers within high-dimensional brain networks over space and time. To the best of our knowledge, this is the first work to utilize LLMs for graph learning in the context of spatio-temporal neurodegenerative disease progression in the brain. Our key contributions are:

- We propose a framework to construct a long-term continuous disease progression trajectory from irregular snapshots while performing graph learning for the constructed long-term series guided by a mixture of different cutting-edge LLMs.

- Compared with classic mechanistic models of neuropathology spread, our model combines multiple mechanisms from the literature to provide higher prediction accuracy with interpretability about how different factors affect the disease progression. The LLMs are capable of suggesting new mechanisms.

- Compared to purely data-driven methods, our approach achieves faster and more stable convergence with improved identifiability; faster and more stable convergence, by incorporating LLMs as constraints in graph learning for spatio-temporal data,

## 2 METHODOLOGY

### 2.1 PROBLEM STATEMENT

Assume for each subject $i$, the corresponding observation is $\tilde{c}_{i,j}^d$, where $j$ is the $j$-th scan of this subject and $d$ is the $d$-th biomarker. Each subject has at least one scan, but the total number varies from subject to subject. Each scan produces a set of D biomarkers, which is common to all scans of all subjects. Our aims are:

- To construct the long-term cohort-level disease progression trajectory, starting from the very early pathology onset time to the late disease stage, from the snapshots of individual-level

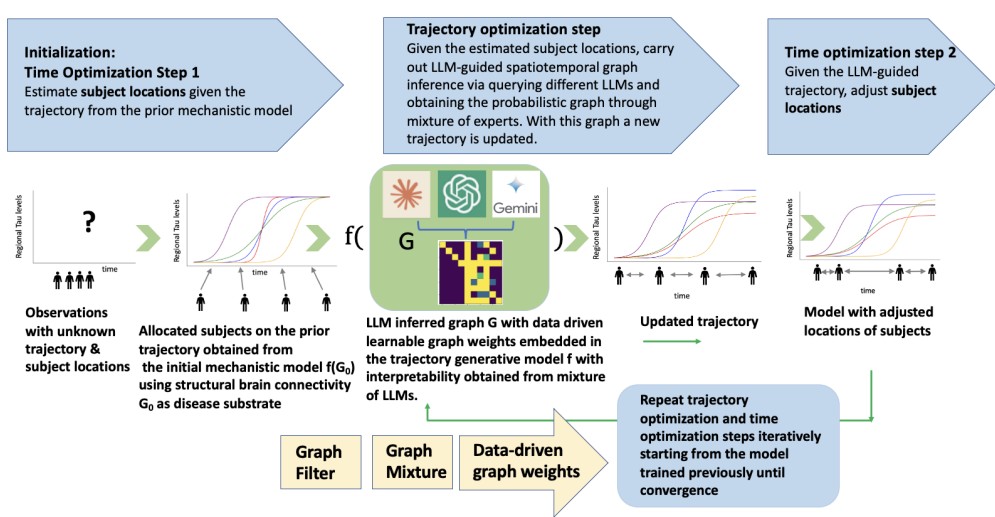

Figure 1: **Model Overview** The proposed framework for constructing a full disease progression process from snapshots, by iteratively estimating subject locations and the embedded graph. The graph plays a dominant role in shaping the disease trajectory. Graph inference includes LLM query, graph filtering, a mixture of expert graphs and data-driven graph weights learning.

observations. To do this, both the relative location of each individual on the cohort-level trajectory and the trajectory itself $\mathbf{c}(t) = f_{\mathbf{G}}(t)$ needs to be estimated dynamically.

- To identify how the biomarkers interact with each other spatially and temporally, by identifying a graph $\mathbf{G}$. A graph element $\mathbf{G}_{k,p}$ indicates the extent to which a biomarker from brain region $k$ is likely to affect the biomarker in brain region $p$. Thus $\mathbf{c}(t) = f_{\mathbf{G}}(t)$. We consider this graph identification in two distinct scenarios, namely:
  - The graph estimation for mechanistic models where the structure of $f_{\mathbf{G}}(t)$ is predefined according to specific physical processes, thus only the graph for how regional biomarkers interact with each other needs to be optimized;
  - The graph estimation for purely data-driven graph learning algorithms where both the graph $\mathbf{G}$ and the structure of $f_{\mathbf{G}}(t)$ are unknown.

## 2.2 OVERVIEW OF THE PROPOSED FRAMEWORK

Figure 1 shows the overview of the framework. We jointly reconstruct the trajectory of disease biomarkers and obtain the relative locations of each individual on the progression time axis. To estimate the trajectory $\mathbf{c}(t) = f_{\mathbf{G}}(t)$, we need to obtain the graph $\mathbf{G}$ and estimate the parameters $\theta$ for the trajectory $f$. To address the identifiability problem by narrowing the collection of possible graphs, to enhance robustness of inference, and to improve interpretability, we propose the following strategy: first, we obtain the initial graph $\mathbf{G}$ through querying the large language model (LLM); then we process $\mathbf{G}$ by graph filtering, graph mixture and then refine the weights of the non-zero elements of the graph in a data-driven way. We then input the graph $\mathbf{G}$ into the generative model to estimate $f_{\mathbf{G}}(t)$; finally we compare the output to real observations and obtain the prediction error.

Initially, neither the trajectory shape nor the relative location of each individual on the trajectory is known. Thus we apply the dual optimization strategy to iteratively optimize the trajectory shape given relative locations of the subject (**trajectory optimization step**) and to optimize the subject locations given the current trajectory shape (**time optimization step**). The model estimation consists of the following steps:

1. The optimization starts with prior knowledge about the trajectory, simulated by the network diffusion model from Raj et al. (2012a). Given this prior trajectory, each subject $i$ can be

allocated to the most appropriate location on the temporal axis within the optimisation of the pseudo time $t_i$.

2. Given the obtained subject location, now the trajectory can be further optimized by querying the graph $\mathbf{G}$ from the LLM as a substrate which produces the trajectory via the propagation mechanism. We simultaneously estimate the parameters $\theta$ for the trajectory $f$. As a result, a new trajectory can be obtained.

3. Given the new trajectory, the subject locations relative to the time axis are further adjusted.

4. Steps 2 and 3 are repeated starting from the model trained previously until convergence.

## 2.3 DUAL OPTIMIZATION IN DISEASE PROGRESSION MODELLING

The existing mechanistic model or the data-driven graph learning models, by default, assume that the subject location on the temporal axis (the observed time) is known. However, in reality we are not able to obtain long-term observations with known observed time from the disease onset. Thus, we need to optimize the trajectory as well as the relative location of each individual on the cohort-level trajectory through the below dual optimization.

### 2.3.1 TIME OPTIMIZATION STEP

For the subject $i$ with observations $\tilde{c}_{ij}, j = 1, .., M_i$, the time gap $\delta_{ij}$ (in years) between the baseline scan of tau-PET to the $j$th follow-up scans are given in the dataset. However, the time from the disease onset to the baseline scan is unknown. Thus we need to estimate such time $t_i^{onset}$. Then $t_{ij} = t_i^{onset} + \delta_{ij}$. This time parameterization enforces the relevant locations among all scans fixed by given $\delta_{ij}$. Thus we define the loss as the sum of squares error (SSE):

$$\mathcal{L}_{(\delta_{ij}, \theta)}\left(t_i^{onset}\right) = \sum_{i=1}^{N} \sum_{j=1}^{M_i} \|\tilde{c}(t_{ij}) - c(t_{ij})\|^2 \tag{1}$$

### 2.3.2 TRAJECTORY OPTIMIZATION STEP

When optimising the trajectory, the relative locations of each subject are temporarily fixed, and then the parameter set of the trajectory is optimised according to

$$\mathcal{L}_{(\delta_{ij}, t_i^{onset})}(\theta) = \sum_{i=1}^{N} \sum_{j=1}^{M_i} \|\tilde{c}(t_{ij}) - c(t_{ij})\|^2 \tag{2}$$

where

$$c(t_{ij}) = \int_0^{t_{ij}} F_{\mathbf{G}}(\theta)d\tau, \quad c(0) = c_0 \tag{3}$$

## 2.4 LLM-GUIDED GRAPH CONSTRUCTION

### 2.4.1 QUERYING A PROBABILISTIC GRAPH FROM LLMS

We aim to uncover the mechanism by which the regional biomarkers spatially interact within the brain's network and change over time by constructing a probabilistic graph. This graph encodes a connection strength level between 0 and 1, indicating whether the biomarker from brain region $k$ influences the biomarker in the brain region $p$. Our prompt strategy is: for a given list of brain regions of interest (ROIs), we query the LLM for every specific region. Specifically, for a given region, we query the LLM as to which other regions in the list are likely to have interactions that can facilitate the progression of diseases. We request each LLM to return for each region a vector containing probabilistic values indicating the connection strength in the existence of the causal relations, together with the reasons for interpretability. Please refer to Appendix A.7 for the detailed prompt. We first prompt the LLM to consider that the neurodegeneration process can be driven by the mixture of different brain connectivities, then explicitly ask the LLM to consider factors of the structural connectome, functional interaction, morphological similarity, geodesic proximity and the microstructural profile covariance, which have been shown to be helpful in disease modelling

Thompson et al. (2024). We define this as the "5-factor" prompt. We query each LLM about which regions are related to a given region in terms of pathology appearance and progression of the regional tau in human brains. To make the result more robust, we request the response 3 times with a temperature of 0.25 and obtain a probabilistic graph by averaging the 3 answers. We obtained the binary mask by thresholding the probabilistic graph.

Meanwhile, these connectomes are available in the Microstructure-Informed Connectomics Database Royer et al. (2022), which can be used as baseline models and verification tools. To ensure that the LLM queried coupled-mechanisms graph is reliable, we compare the graph with the summation of the five types of brain connectivities from this database, each filtered to varying degrees, as displayed in Appendix A.3.

### 2.4.2 GRAPH FILTERING

For each graph, we threshold the measure of connectivity to find a binary graph. To retain significant interactions in the dynamical system and minimise the number of learnable variables in the graph to avoid identifiability and overfitting problems, we apply thresholding to the LLM graphs by only keeping strength levels only above a specific threshold. We define the highest threshold that can retain the performance as the "the critical threshold", and the corresponding minimum number of edges to retain the model performance "the critical edge number", defined as $N^*_{edge}$. To make the comparison between the disease spreading model using the LLM-guided graph and the traditional structural connectome, we also filter the structural connectome so that the number of non-zero elements in the connectome is the same as that in the filtered LLM graph.

### 2.4.3 MIXTURE OF GRAPHS FROM DIFFERENT LLM EXPERTS

To benefit from the expertise of different LLMs and increase the robustness of the obtained graph, we mix the graphs from different LLMs using a weighted sum. Since the performance of different LLMs varies in this specific task, we propose a way to combine the graphs from different LLMs based on their individual performance. Specifically, we choose $M$ LLMs and record the critical edge number $N^*_{\text{edge}_i}$ for the $i$th LLM. We define the weights assigned to each LLM to be inversely proportional to their critical edge number, since graphs with a lower $N^*_{edge}$ are more robust:

$$w_i = \frac{\left(1/N^*_{\text{edge}_i}\right)^\alpha}{\sum_{j=1}^M \left(1/N^*_{\text{edge}_j}\right)^\alpha} \tag{4}$$

The parameter $\alpha$ controls the emphasis on the models with lower critical edge numbers.

### 2.5 EMBEDDING THE LLM-GUIDED GRAPH INTO THE DYNAMICAL SYSTEM FOR TRAJECTORY CONSTRUCTION

Next, we embed the graph $\mathbf{G}$ into the generative model $f_{\mathbf{G}}(t)$ and compare this model's output with actual observations to determine the prediction error. Through this process, $\mathbf{G}$ is shaped by integrating both the expert insights from LLM and the capabilities of data-driven analysis. We explore the identification of this graph through two specific case studies: 1) Graph estimation for mechanistic models where the function $f_{\mathbf{G}}(t)$'s structure is pre-determined by experts during the model's development. In this case study, the focus is solely on optimizing the graph, which details interactions between regional biomarkers; 2) Graph estimation for data-driven graph learning algorithms where both the graph $\mathbf{G}$ and the function structure $f_{\mathbf{G}}(t)$ are initially unknown. We use the propagation of the tau protein on the brain graph as a case study.

### 2.5.1 GRAPH LEARNING FOR MECHANISTIC MODELS

**Model 1 (baseline) - Spreading model for regional tau on the weighted structural connectome**
The baseline mechanistic model for describing regional tau interactions in the brain is:

$$\frac{d\mathbf{c}}{dt} = -k[\mathbf{L}\mathbf{c}(t)] + \alpha\mathbf{c}(t) \odot [v\mathbf{p} - \mathbf{c}(t)] \tag{5}$$

The first term describes the diffusive spread of pathology between connected brain regions. The graph $\mathbf{L}$ is the Laplacian of the structural connectivity matrix $\mathbf{A}$, defined as $\mathbf{L} = \mathbf{D} - \mathbf{A}$, where each element $D_{i,i}$ of the diagonal degree matrix $\mathbf{D}$ is the sum of the weights of the edges connected to vertex $i$. $\mathbf{L}$ is normalized by the row summation following Raj et al. (2012a). Three learnable parameters $k$, $\alpha$ and $v$ represent the rate of pathology spreading, pathology aggregation and the general level of convergence. The second term describes the production of pathology in each node, up to a regionally varying carrying capacity $\mathbf{p}$, following Chaggar et al. (2023). This is calculated from the 99th percentile of the tau distribution at each region. However, this mechanistic model makes two assumptions, which may oversimplify the disease process: i) the spreading of tau only relies on the weighted structural connectome; ii) the propagation of tau in disease progression is not affected by other biomarkers. We aim to address these limitations by defining the following models using our proposed framework.

**Model 2 (baseline) - Coupled-mechanisms of Tau spreading via a simple linear mixture of connectomes** We apply a mixture of connectomes model proposed byThompson et al. (2024) on the long-term cohort-level disease progression model, defined as

$$\frac{d\mathbf{c}}{dt} = -k[(w_1\mathbf{L_1} + w_2\mathbf{L_2} + w_3\mathbf{L_3} + w_4\mathbf{L_4} + w_5\mathbf{L_5})\mathbf{c}(t)] + \alpha\mathbf{c}(t) \odot [v\mathbf{p} - \mathbf{c}(t)] \tag{6}$$

where $\mathbf{L_1}$, $\mathbf{L_2}$, $\mathbf{L_3}$, $\mathbf{L_4}$ and $\mathbf{L_5}$ represent the graph Laplacian matrices obtained from the structural connectome, functional connectome, morphological similarity matrix, geodesic proximity and microstructural connectome respectively. We define the brain regions according to the Desikan-Killiany Atlas Desikan et al. (2006). By considering linear combinations of different graphs, more mechanisms for tau propagation are considered. However, this method only considers the simplest way of combining mechanisms, with limited interpretability of how the interaction occurs at each region and how such processes differ across brain regions.

**Model 3 (proposed) - Coupled-mechanisms of Tau spreading via a complex mixture of connectomes queried from LLM**

$$\frac{d\mathbf{c}}{dt} = -k[\mathbf{L_{LLM}}\mathbf{c}(t)] + \alpha\mathbf{c}(t) \odot [v\mathbf{p} - \mathbf{c}(t)] \tag{7}$$

where the graph Laplacian , $\mathbf{L_{LLM}}$, is obtained using the diagonal degree matrix $\mathbf{D_{LLM}}$ and the adjacent matrix $\mathbf{A_{LLM}}$:

$$\mathbf{L_{LLM}} = \mathbf{D_{LLM}} - \mathbf{A_{LLM}} \tag{8}$$

and

$$\mathbf{A_{LLM}} = \mathbf{G_{LLM}} \odot \mathbf{W} \tag{9}$$

Rather than directly inputting a graph Laplacian calculated from a known weighted structural connectome, we query the probabilistic graph $\mathbf{G_{LLM}}$ from an LLM and then calculate the graph Laplacian $\mathbf{L}_{LLM}$ using the proposed prompt, as defined in Appendix A.7., where we ask the LLM to consider the mixture of biological factors not limited to structural connections, but also other related brain graphs such as functional connectome and geodesic proximity etc. Thus, this not only takes into account the diffusion process along the white matter bundles but also considers other factors regarding the tau accumulation from the literature in the knowledge base of different LLMs, including Claude3.5, GPT4-turbo and Google Gemini 1.5 Pro. After obtaining the filtered graph $\mathbf{G_{LLM}}$, we learn the weights of the non-zero elements of the graph $\mathbf{W}$ in a data-driven way during the model training. $\mathbf{W}_{i,j}$ represents the extent of interaction between regions $j$ and $i$. We carry out the same procedure for the graphs in the baseline model for a fair comparison.

### 2.5.2 Data-driven graph learning models for continuous, irregular series

Apart from mechanistic models which have a relatively fixed structure, there also exist pure data-driven methods for inferencing a graph structure of how variables interact from time series data, where both the graph and the structure of the model are unknown. Due to the huge extent of flexibility, the identifiability of the graph needs to be considered through various regularization methods. However, those methods are robust especially when the size of the graph is small, and the data is relatively perfect (such as synthetic data). For our problem setting, where the dimension of the brain is relatively high and the data is complex and noisy, the robustness and stable convergence of such methods are hard to guarantee. One state-of-the-art data-driven graph learning algorithm is the Neural Graphical

Model: NGM Bellot et al. (2021), a score-based learning algorithm based on penalized Neural Ordinary Differential equations, which is applicable to the general setting of irregularly-sampled multivariate time series. The derivative of the $j$-th variable in the dynamical system $f_j(\mathbf{C})$ is defined by stacking several layers of the neural network.

$$f_j(\mathbf{X}) := \phi\left(\cdots\phi\left(\phi\left(\mathbf{X}A_1^j\right)A_2^j\right)\cdots\right)A_M^j \tag{10}$$

where $\mathbf{x} = (x_1, \ldots, x_d)$ contains $d$ distinct stochastic processes of regional disease dynamics and $\mathbf{X} \in \mathbb{R}^{n \times d}$ is is the sequence of $n$ $d$-dimensional instantiations of $\mathbf{x}$. $\phi(\cdot)$ is the activation function. The graph is obtained by penalizing the weight of the first layer $A_1^j$. Specifically, enforcing the kth column of $\|[A_1^j]_{\cdot k}\|_2 = 0$ will eliminate the local dependence of the $j$-th stochastic process on the $k$-th stochastic process. The Group Lasso (Zhao & Yu (2006)) and adaptive Group Lasso (AGL, Zou (2006)) methods have been used for the regularization purpose, with $\lambda_{GL}$ and $\lambda_{AGL}$ as the regularization strengths respectively, where the weights of AGL are based on GL. See Appendix A.2 for definitions. With the structure of Neural ODE, the model can provide continuous modelling and handle irregularly sampled data. However, the model needs to be carefully contained to guarantee identifiability, which is challenging for high-dimensional data.

In order to constrain the graph in NGM, we set the corresponding $\|[A_1^j]_{\cdot k}\|_2 = 0$ to be 0 according to the zero elements of the graph we queried from the expert graphs such as the LLM graph from Claude3.5, and the corresponding parameters which are not masked will be estimated. During the process, the GL method can be optionally applied to provide further regularization. The constraints from the LLM provide an interpretable and more stable optimization process for graph learning while constructing the continuous trajectory from neural ODE.

## 3 EXPERIMENTS AND RESULTS

In this section, we demonstrate that our proposed LLM-guided graph improves prediction accuracy in disease progression modeling for both mechanistic models (where the model structure and physical processes are known, but parameters are not) and data-driven spatiotemporal graphical models (where the entire model structure is unknown). Additionally, it offers better identifiability and interpretability.

For mechanistic modelling, we compare three approaches: the progression model using only the structural connectome (model 1), a linear combination of brain connectivity modalities (model 2) Thompson et al. (2024), and the progression model embedded with our LLM-guided graph (model 3). To ensure fairness, all models use the same graph filtering methods, and their weights are learned in a consistent, data-driven manner. We show that when the number of learnable parameters exceeds what is necessary, various graphs can achieve similar accuracy, though identifiability suffers. By sparsifying the graph through thresholding, our LLM-guided graph constrained by coupled mechanisms outperforms alternatives with fewer parameters.

For data-driven methods, we compare the NGM's initial two-step Lasso method with the NGM constrained by our LLM-guided graphs. Results indicate that neural ODEs constrained by LLM graphs achieve faster convergence and higher accuracy than other regularization techniques. Additionally, synthetic experiments in Appendix 6 illustrate that our method achieves higher accuracy in graph inference compared to existing spatiotemporal modelling methods when the ground truth is known.

### 3.1 TRAINING METHODOLOGY

We analyze Tau dynamics using a cohort of 255 individuals from the Alzheimer's Disease Neuroimaging Initiative (ADNI). See Appendix A.4 for a detailed description. The subjects used for model training, test and validation have 1-4 scans with altogether 378 observations. We implement 3-fold cross-validation by randomly assigning 35 subjects each to validation and test sets, with the remaining subjects forming the training set. All longitudinal scans from the same subject are kept together in their assigned sets, preserving the actual time intervals between measurements. The validation step happens after an epoch of trajectory optimization on the training data, i.e. the subjects from the validation set are allocated on the trajectory from each training epoch through stage optimization. The training of the trajectory will be stopped if the performance on the validation set ceases to improve. Finally, the relative location of the subjects from the test set is estimated, and the corresponding model performance is recorded as the test metric.

Table 1: **Mechanistic Model Comparison with different graph embedded**

| Model Name | N_edges | Test SSE | Test Pearson R | Test AIC |
|---|---|---|---|---|
| Claude 3.5 Sonnet | 314 | 14.05 ± 1.33 | 0.66 ± 0.03 | 541.49 ± 14.61 |
| Structural Connectome | 314 | 24.89 ± 2.36 | 0.36 ± 0.03 | 576.96 ± 12.51 |
| Gpt4-turbo | 650 | 13.17 ± 1.65 | 0.70 ± 0.02 | 1209.27 ± 17.19 |
| Structural Connectome | 650 | 13.38 ± 1.86 | 0.68 ± 0.01 | 1210.12 ± 18.12 |
| Gemini Pro 1.5 | 396 | 14.09 ± 1.73 | 0.67 ± 0.02 | 705.47 ± 16.89 |
| Structural Connectome | 396 | 19.10 ± 1.75 | 0.51 ± 0.02 | 724.59 ± 12.85 |
| Mixed LLMs | **284** | 14.27 ± 1.41 | 0.64 ± 0.03 | **482.40 ± 15.21** |
| Linearly-mixed Connectomes | 314 | 22.50 ± 3.02 | 0.46 ± 0.02 | 570.40 ± 15.61 |

## 3.2 COMPARISON OF MECHANISTIC MODELS

We first use the mechanistic models to capture the propagation of tau (a key biomarker in Alzheimer's disease) among brain regions. A detailed description of the dataset can be found in the supplementary materials. We optimize the mechanistic models defined previously in section 2.5.1. For each type of graph, we threshold a measure of connectivity to find a binary graph. Low thresholds (low sparsity) lack identifiability as they contain many redundant paths that support pathology propagation. High thresholds (high sparsity) capture only the important connections, but as the threshold increases experience catastrophic failure once strongly connected brain regions are fully severed. In general, we seek the sparsest graph (maximising interpretability) that is able to recover the pathology propagation pattern. This allows us to avoid over-fitting.

We estimate weights for filtered binary 5-factor-prompt LLM graphs by converting their non-zero elements into positive learnable parameters. Following the mixture of experts method (section 2.4.3), we combine graphs from Claude 3.5 and Gemini 1.5 Pro with weights 0.865 and 0.135 respectively. GPT4's graph was excluded due to its higher edge count, despite better performance at higher densities. We apply identical filtering and weight estimation to structural brain connectivity and their linear combinations. Table 1 compares model performance across different graph substrates on the test set. The LLM-derived graphs achieve superior predictions with fewer learnable variables compared to single or linearly combined connectivity models. Figure 2 plots Pearson R correlation and AIC against parameter count. Dense graphs show poor identifiability due to similar performance across types. However, the sparse LLM graph demonstrates superior fitting, leading to our final data-driven weighted mixed LLM graph. The visualization of the predicted tau progression pattern versus real observation can be found in Appendix A.5.

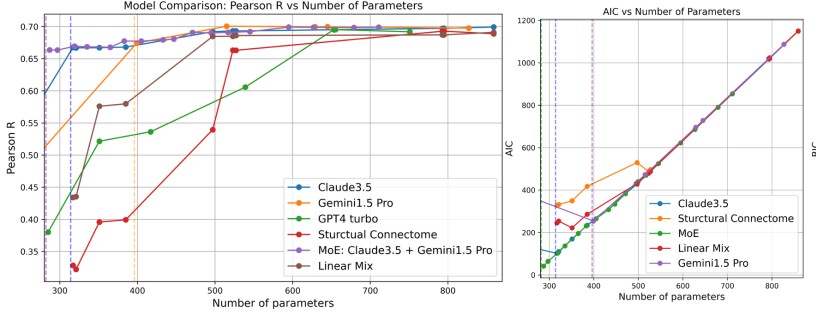

Figure 2: **Model Performance: R correlation on test set vs parameter number (Left); AIC on training set vs parameter number (Right)**. The dashed vertical lines represent the critical edge numbers of LLMs. The graph obtained from the mixture of LLMs provides the lowest AIC at the smallest parameter number, followed by Claude 3.5. As the number of learnable parameters increases, all models tend to have the same performance level. The LLM-based graphs allow the model to retain high performance to much greater sparsity levels than the connectivity-based graphs.

### 3.3 COMPARISON OF DATA-DRIVEN GRAPH LEARNING MODELS IN TIME SERIES

Section 2.5.1 considers traditional mechanistic models that explicitly define the graph's contribution to physical processes. We now examine the Neural Graphical Model (NGM) Bellot et al. (2021), a data-driven approach where the model structure is unknown. For such models, graph regularization must be carefully designed to constrain the optimization space and improve convergence, particularly with high-dimensional, noisy data. The original NGM uses a two-step Group Lasso (GL) and Adaptive Group Lasso (AGL) regularization, controlled by hyperparameters $\lambda_{GL}$ and $\lambda_{AGL}$ (detailed in Appendix A.2). However, this approach yields unstable graph inference in our case study - figure 3.3 demonstrates that two separate runs with identical data and hyperparameters produce different graphs. Table 2 of three-fold cross-validation shows that using the LLM-derived expert graph as a constraint improves both time series fitting accuracy and algorithmic convergence. As outlined in section 2.5.2, we enhance the original regularization by either 1) using only the sparse LLM graph ($\lambda_{GL} = 0$, $\lambda_{AGL} = 0$) or 2) combining a denser LLM graph with Group Lasso ($\lambda_{GL} = >0$, $\lambda_{AGL} = 0$), where $N_{Raw}$ is the number of edges in the LLM graph when starting the algorithm, while $EdgeNumber$ is the remained edge after algorithm convergence, indicating the effective number of learnable parameters needed. Both approaches achieve higher accuracy with significantly fewer edges, indicating that LLM effectively captures key disease transmission pathways. Appendix A.2 demonstrates that these graph-constrained models converge faster and more stably.

Table 2: **Model Comparison for data-driven graph learning in neural dynamical system**

| Model | Test SSE | Edge Number |
|---|---|---|
| NGM$_{AGL}$ ($\lambda_{GL}$ = 0.1, $\lambda_{AGL}$ = 0.10) | 13.67 ±2.81 | 348 ± 57 |
| NGM$_{AGL}$ ($\lambda_{GL}$ = 0.1, $\lambda_{AGL}$ = 0.05) | 13.63 ± 2.82 | 517 ± 157 |
| NGM$_{AGL}$ ($\lambda_{GL}$ = 0.1, $\lambda_{AGL}$ = 0.01) | 13.74 ± 2.72 | 934 ± 322 |
| NGM$_{Mix-LLM-constrained}$ ($N_{Raw}$ = 310, $\lambda_{GL}$ = 0.) | **13.49** ± 2.89 | 310 |
| NGM$_{Mix-LLM-constrained}$ ($N_{Raw}$ = 448, $\lambda_{GL}$ = 0.1) | 13.66 ± 2.87 | 245 ± 29 |
| NGM$_{Claude3.5-constrained}$ ($N_{Raw}$ = 226, $\lambda_{GL}$ = 0.) | 13.57 ± 2.93 | 226 |
| NGM$_{Calude3.5-constrained}$ ($N_{Raw}$ = 382, $\lambda_{GL}$ = 0.1) | 13.54 ± 2.81 | **211 ±12** |



Figure 3: The plot compares stability of learnt graphs where different graphs are obtained from the NGM without LLM constraint in two separate runs while graphs are more robust from our method.

Apart from NGM, there are other graph-learning methods for time series. We demonstrate in the synthetic data experiments with known ground truth that when the data dimension is high, these methods provide very different graphs, which hints at the problem of graph identifiability. Meanwhile, the LLM-guided graph is similar to the ground truth due to the expert knowledge. Please refer to Appendix A.1 for comparison with other baseline models.

### 3.4 INTERPRETABILITY FROM LLM

Figure 12 displays an example of the output from Claude-3.5-Sonnet, when queried about the disease-related interactions between the left rostral anterior cingulate cortex and other 67 cortical brain regions defined in the Desikan-Killiany Atlas. Apart from the five factors we explicitly provide, we further encourage the LLM to think about other factors that might contribute to disease spread (displayed in red). As a result, factors like the effect of neurotransmitter systems are proposed. This has been quantitatively shown to have a significant influence on the spreading of regional tau by Soskic et al. (2024). Reasoning from other LLMs can be found in the Appendix.

*Reasoning from Claude3.5 Sonnet*

The strength dictionary reflects the potential relationships between ctx_lh_rostralanteriorcingulate and other brain regions in the context of Alzheimer's disease (AD) progression. Here's the rationale for the connections:

**1. Structural connectivity:** The rostral anterior cingulate cortex (rACC) has strong connections with other frontal regions, particularly the medial prefrontal cortex. This is reflected in high strength values for ctx_lh_rostralmiddlefrontal (0.8), ctx_lh_superiorfrontal (0.7), and ctx_lh_medialorbitofrontal (0.7).

**2. Similarity of cortical morphology:** Regions with similar cortical structure are more likely to be affected similarly in AD. This is why other cingulate regions like ctx_lh_caudalanteriorcingulate (0.8) and ctx_lh_posteriorcingulate (0.6) have high strength values.

**3. Spatial proximity:** Nearby regions are more likely to be affected due to the spread of pathology. This explains the high values for adjacent frontal regions like ctx_lh_frontalpole (0.6) and ctx_lh_parsorbitalis (0.6).

**4. Functional connectivity:** The rACC is part of the default mode network (DMN), which is heavily implicated in AD. Other DMN regions like ctx_lh_precuneus (0.4) and ctx_lh_inferiorparietal (0.3) have moderate strength values.

**5. Microstructural profile covariance:** Regions with similar cytoarchitecture may be similarly vulnerable to AD pathology. This contributes to the high values for other prefrontal regions.

**6. Interhemispheric connections:** The corresponding region in the right hemisphere, ctx_rh_rostralanteriorcingulate (0.7), has a high strength value due to interhemispheric connections.

**7. Known AD progression patterns:** Regions known to be affected early in AD, such as the entorhinal cortex (ctx_lh_entorhinal, 0.3) and parahippocampal gyrus (ctx_lh_parahippocampal, 0.3), have moderate strength values due to their role in disease progression.

**8. Neurotransmitter systems:** The rACC is involved in cholinergic and dopaminergic systems, which are affected in AD. This influences connections with regions like the insula (ctx_lh_insula, 0.5) and other prefrontal areas. Regions with low strength values (0-0.2) are those with limited known direct connections or involvement in AD pathology related to the rACC, such as primary sensory and motor areas. The values reflect a combination of direct and indirect connections, considering both anatomical and functional relationships in the context of AD progression.

Figure 4: This figure displays one representative example of an output from Claude 3.5. Factors in red (6 - 10) are those which weren't mentioned in the prompt.

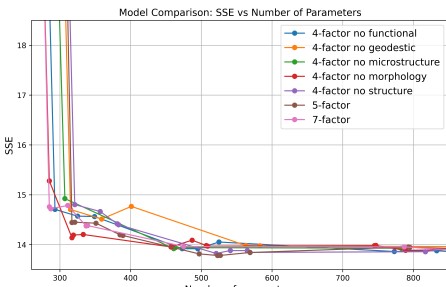

Figure 5: **Ablation study of different prompts - SSE vs number of remaining edges**.

## 3.5 ABLATION STUDY OF PROMPT COMPONENTS

Figure 5 displays the performance of the best LLM, Claude 3.5, across the different prompts. We consider the original 5-factor prompt; removing each different factor from the original prompt (4-factor prompts); as well as the 7-factor prompt, where two more factors (neurotransmitter density as suggested by Soskic et al. (2024) and metabolic correlation map as suggested by Adams et al. (2019)) have been added. The 5-factor prompt provides the lowest overall test SSE while removing the geodesic proximity significantly decreases the accuracy. The "7-factor" prompt offers a way of extending the knowledge outside the existing five connectomes that are available in the MICA-MICS database by explicitly adding two more features to the prompt. This prompt has further decreased the critical edge number compared with the 5-factor prompt.

## 4 CONCLUSIONS

We propose a novel framework designed to construct long-term continuous disease progression trajectories from irregular snapshots while simultaneously performing graph learning on the generated long-term series. By coupling multiple mechanisms from LLMs, our model surpasses the classic mechanistic model, delivering higher prediction accuracy. Furthermore, by integrating LLMs as constraints in data-driven graph learning methods for time series, our approach not only accelerates and stabilizes convergence but also enhances identifiability and interpretability. For future work, we will look at other indicators of neurodegeneration other than tau. And we will do more exploration to increase LLM performance. This framework can be easily adapted to other domains since the expertise comes from LLM rather than any specific knowledge base.

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

# A  APPENDIX

**Summary of Appendices.**

- Evaluation of learnt graph on synthetic data
- More results for NGM modelling
- Verification of the LLM graph from the disentangled bran graphs
- Data Description
- Disease Progression Visualization
- More interpretation from LLMs
- Prompts
- Related work

## A.1  EVALUATION OF LEARNT GRAPH ON SYNTHETIC DATA

Apart from NGM discussed previously, there are other graph learning methods for time series which do not explicitly aim to generate time series as the methods before. Instead, they focus more on discovering the graph of how different variables interact with each other from time series. Thus, they can also be baselines to compare with our proposed method on the accuracy of graph inference. The Structural Vector Autoregression Model (SVAM) (Hyvärinen et al., 2010), an extension of the LiNGAM algorithm to time series, is another representative model. PCMCI (Runge et al., 2017), a representative independence-based approach to structure learning with time series data, extends the PC algorithm. Another method, Dynamic Causal Modelling (DCM), is a representative two-stage collocation approach in which derivatives are first estimated on interpolations of the data, and a penalized neural network is learned to infer G (extending the linear models of Ramsay et al., 2007; Wu et al., 2014; Brunton et al., 2016). However, the optimization goal is to minimize the modeled derivatives with the interpolated derivative of the data, rather than directly optimizing the trajectory

itself. Thus, for those methods, we cannot compare the modelling accuracy of disease progression, but instead, we evaluate the obtained graph with the ground truth graph from synthetic data.

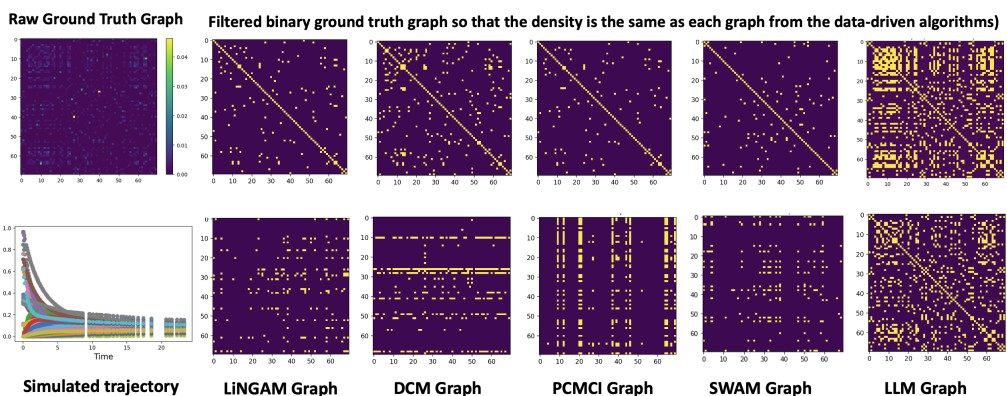

Figure 6: **Synthetic Data Experiments for graph comparison with the ground truth** This figure displays the graphs obtained from different data-driven graph learning methods compared with the filtered ground truth graph at the same density.

To more directly evaluate the graph derived from our proposed algorithm and compare it with more algorithms of learning the graph from time series which are unable to construct disease progression trajectory due to their incapabilities of handling continuous irregular data, we generated synthetic data for the comparison purpose where the ground truth of the graph is known and can be queried from LLM. Specifically, we simulate the air pollution of 70 main cities in China by creating a graph of spatial proximity using the inverse of the geodesic distance calculated from the coordinates of each city. Then we simulate the air pollution by using the one-component diffusion process on the proximity network Raj et al. (2012a), i.e. assuming that the pollution diffuses from the cities of the high concentration of pollution to the rest of the cities, eventually reaching the status that all the cities have the equivalent concentration with time going by. We apply different graph inference methods to the simulated time series data and compare the obtained graph in Figure 6. It can be observed that LLM can capture the main patterns of the relations while other methods struggle to capture many existing connections.

## A.2 MORE RESULTS FOR NGM MODELLING

As defined in Bellot et al. (2021); Zou (2006); Zhao & Yu (2006), the definition of the GL and AGL regularization are:

$$
\rho_{\mathrm{GL}}\left(\mathbf{f}_\theta\right) := \lambda_{\mathrm{GL}} \sum_{k,j=1}^d \left\| \left[A_1^j\right]_{\cdot k} \right\|_2, \quad \rho_{\mathrm{AGL}}\left(\mathbf{f}_\theta\right) := \lambda_{\mathrm{AGL}} \sum_{k,j=1}^d \frac{1}{\left\| \left[\hat{A}_1^j\right]_{\cdot k} \right\|_2^\gamma} \left\| \left[A_1^j\right]_{\cdot k} \right\|_2 \quad (11)
$$

where $\hat{A}_i^j$ is the GL estimate. The parameters $\lambda_{GL}$ and $\lambda_{AGL}$ control the regularization intensity. Additionally, $\gamma > 0$ and $\| \cdot \|_2$ represent the Euclidean norm. **AGL** utilizes its base estimator to provide a preliminary, data-driven estimate, allowing it to shrink groups of parameters with different regularization strengths.

Figure 7 compares the converge plots of the AGL-constrained NGM and the proposed LLM graph-constrained NGM by displaying the SSE vs the number of iterations. Since the formulation of the regularization of AGL is dependent on the weight from GL, the total number of iterations needs to be accumulated. The plots on the left demonstrate that the convergence of the GL method is not stable and needs a relatively large number of iterations to be converged namely 1000 runs. Then followed by the AGL method starting from an initial SSE of around 2000, whose convergence stabilizes after 300 iterations with some vibration afterwards. While for the proposed LLM constrained regularization, converge is achieved around 150 iterations in total starting from an initial SSE of around 600. This

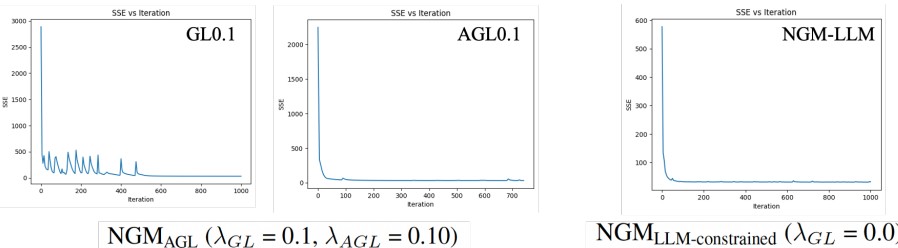

$$\text{NGM}_{\text{AGL}} \ (\lambda_{GL} = 0.1, \ \lambda_{AGL} = 0.10) \qquad \text{NGM}_{\text{LLM-constrained}} \ (\lambda_{GL} = 0.0)$$

Figure 7: **Convergence Plot - Comparison for the AGL regularization and the proposed LLM-constrained regularization**

shows that the proposed LLM graph-constrained method provides a good regularization from the expert knowledge.

### A.3 VERIFICATION OF THE LLM GRAPH FROM THE DISENTANGLED BRAN GRAPHS

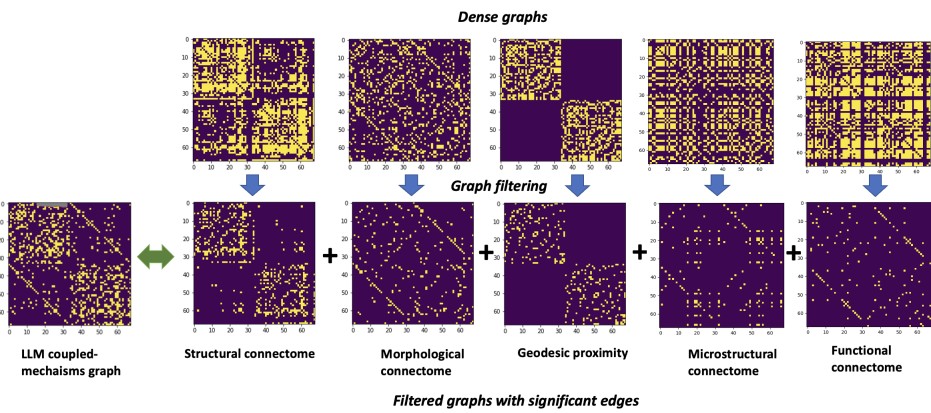

Figure 8: **Verification of the LLM graph** Verification of Large Language Model (LLM) coupled-mechanisms through comparison with disentangled brain connectivity patterns on the right is shown. The dense graphs (top row) undergo filtering to reveal significant edges (bottom row), demonstrating structural similarities between the LLM coupled-mechanisms graph and various brain connectomes. Key similarities include: block-like clusters in top-left and bottom-right regions(matching structural and geodesic patterns), consistent diagonal elements (aligned with functional connectome), sparse central connectivity (similar to morphological patterns), and modular organization. These parallel patterns suggest that LLM mechanisms may mirror fundamental principles of brain connectivity organization across structural, functional, and geodesic dimensions.

We carry out verification of Large Language Model (LLM) coupled-mechanisms through comparison with disentangled brain connectivity patterns on the right, as is shown Figure 8. Key patterns include:

**Block-like Clustering Pattern**: The LLM graph shows distinct block structures in the top-left and bottom-right corners. This pattern is strongly mirrored in the Structural connectome, which also displays similar dense clusters in these regions. The Geodesic proximity graph reinforces this pattern, particularly in the bottom-right quadrant.

**Diagonal Elements**: The LLM graph exhibits scattered diagonal elements across the matrix. This diagonal pattern is particularly visible in the Functional connectome. Similar diagonal structures appear in the Microstructural connectome and Morphological connectome.

**Edge Density Gradients**: The LLM graph shows varying densities of connections, with some areas being more concentrated than others. This gradient pattern is similar to what's observed in the

Structural and Geodesic proximity graphs. The transition between dense and sparse regions follows comparable patterns.

## A.4 DATA

### A.4.1 PET IMAGE PROCESSING

The dynamics of aggregated tau protein are modelled in this study utilizing tau-PET standardized uptake value ratios (SUVRs) obtained from the Alzheimer's Disease Neuroimaging Initiative (ADNI) database (adni.loni.usc.edu) Landau et al. (2021). Off-target binding effects of the radiotracer in subcortical regions necessitate their exclusion from our analysis Groot et al. (2022).

### A.4.2 SUBJECT INCLUSION CRITERIA

The ADNI Tau SUVRs cohort used in this study is characterized by positive amyloid beta status (another key biomarker related to Alzheimer's Disease), where the label has been provided in the dataset already. Then for each cortical region of interest, we implement a two-component Gaussian mixture model on the SUVR measurements from the collective subject pool. The component with the lower mean is identified as representative of the distribution of non-pathological signals, and we establish a cutoff for tau-positivity as the mean plus one standard deviation of this component. As a result, the included subjects are amyloid-positive and tau-positive in at least one region, encompassing subjects across the spectrum from cognitively unimpaired to those with cognitive impairment and dementia. The subjects with amyloid positive but all regions being tau negative can be used as a control group or an alternative way to initialize the disease onset. This selection criterion is predicated on our interest in individuals who are at potential risk of accumulating abnormal tau aggregates. We normalize the tau data for all participants (i = 1, ..., N) to a range between 0 and 1 using the formula: $(tau_i - tau_{min})/(tau_{max} - tau_{min})$ Here, $tau_{\min}$ and $tau_{\max}$ are the minimum and maximum tau values, respectively, determined across all participants and regions, thereby preserving the variance in measurement scales both between subjects and across regions.

### A.4.3 INITIALIZATION FOR DIFFERENTIAL EQUATIONS

Please note that the initialization of the differential equations differs for different methods. For mechanistic modelling, we choose a pair of inferior temporal cortex regions at both hemispheres, which have been discovered to perform best in the cohort-level tau prediction for both unimodal and multimodal connectomes in ADNI by Thompson et al. (2024). While for the data-driven methods, the models, by default, use the observation at the earliest disease stage to start with.

## A.5 DISEASE PROGRESSION VISUALIZATION

Figure 9 visualizes how the proposed framework uses the snapshots of the individual cross-sectional data or the short longitudinal data to construct the full disease progression trajectory. If an individual has longitudinal scans, the real-time gap (in years) between those scans remains. The pseudo-time axis thus reflects the relative disease stage across the subjects. After training, when a new subject arrives, this subject can be allocated to a position on the trajectory (as shown in orange), using the time optimization step described in section 2.3.1. Then the disease stage of this subject relative to the whole cohort can be obtained, which will be useful in diagnosis, as this time demonstrates the extent of the pathology progression. Since the trajectory is generated using the LLM-guided graph as the substrate for pathology progression, we can understand how different brain regions interact with each other from the graph as well as the corresponding reasoning from LLM.

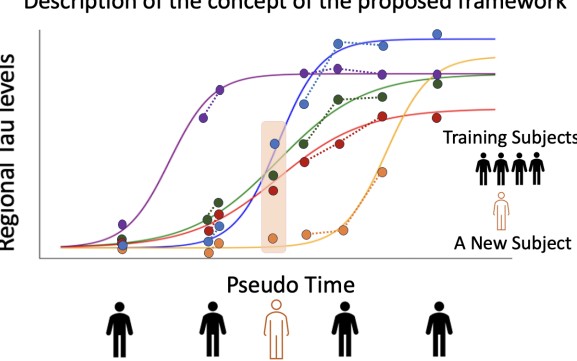

Figure 9: **Description of the concept of the proposed Framework** This figure visualizes how the proposed framework uses the snapshots of the individual cross-sectional data or the short longitudinal data to construct the full disease progression trajectory. Each colour represents one brain region. The dots represent real observations. The dots connected with dashed lines represent the longitudinal observations from the same subject, where the real-time gap between the scans is available in the dataset and thus remains. The curves represent the model fitting.

Below, we display the disease progression pattern of tau from the real observations vs the fitting using the best mechanistic model guided by the mixture of LLMs constructed using our proposed framework via brain mapping relative to the pseudo time axis. After allocating all the subjects on the pseudo-time axis, using the method shown by Figure 9, the observations and the model fitting at the relative locations of 0, 1/4, 2/4, 3/4, 4/4 are visualized (if there is no observation at the exact point, the closest observation nearby is taken, and the modelling fitting is taken at the same nearby time). The colour bar, shared by all brain plots, displays the level of normalized tau SUVR (described in section A.4.2) at each brain region. It can be observed that the major patterns of tau distribution with time have been captured using our proposed model.

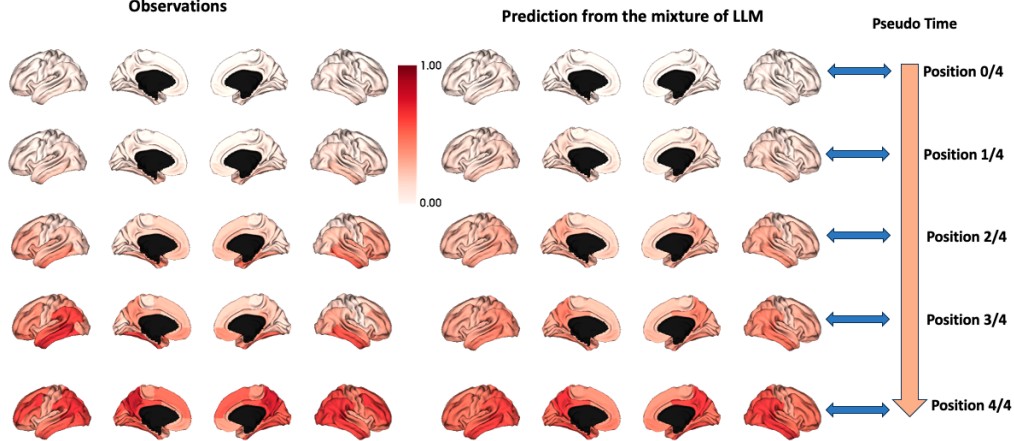

Figure 10: The plot displays the tau distribution pattern during disease progression from the real observations vs the fitting using the best mechanistic model guided by the mixture of LLMs constructed using our proposed framework via brain mapping relative to the pseudo time axis. The colour bar, shared by all brain plots, displays the level of normalized tau SUVR at each brain region.

A.6    MORE INTERPRETATION FROM LLMS

Below, we display the reasoning of the same region from the rest of the LLMs, where it can be observed that GPT4-turbo provides the least analysis. This might be one of the reasons that GPT4-turbo performs the worst among the three language models.

> **Reasoning from Gemini 1.5Pro**
> The rostral anterior cingulate cortex (ACC) plays a crucial role in the default mode network (DMN) and salience network (SN), which are known to be affected in Alzheimer's disease. Thus, regions within these networks, such as the posterior cingulate, medial prefrontal cortex, and insula, are assigned high strength due to their **functional connectivity**. **Structural connectivity** is considered by assigning moderate to strong strength to regions directly connected to the rostral ACC, like the caudal anterior cingulate and rostral middle frontal areas. **Spatial proximity** is factored in by giving higher strength to regions closer to the rostral ACC. **Similarity in cortical morphology** is considered, with regions sharing similar cytoarchitecture having moderate strength. **Microstructural profile covariance** is implicitly considered through the functional and structural connections. **Other factors like the progression pattern of tau pathology** are considered by giving higher strength to regions known to be affected early in the disease course, such as the entorhinal cortex and hippocampus. Less likely connections, like **visual processing areas,** are assigned lower strength.

Figure 11: This figure displays one representative example of an output from Gemini 1.5 Pro.

> **Reasoning from GPT4-turbo**
> The connection strengths are based on various factors such as **structural connectivity, similarity of cortical morphology, spatial proximity, functional connectivity, and microstructural profile covariance**. Regions like the entorhinal cortex and parahippocampal cortex show very strong connections due to their roles in **memory and spatial navigation**, which are heavily impacted in Alzheimer's disease. The medial orbitofrontal cortex and posterior cingulate also show strong connections due to their involvement in **cognitive functions and emotional processing**. Lesser connected regions like the cuneus and pericalcarine are less involved in the primary pathology of Alzheimer's disease.

Figure 12: This figure displays one representative example of an output from GPT4 Turbo.

A.7    PROMPTS

Listing 1: Prompt for regional tau graph

```
SYSTEM_PROMPT = f'''You are a knowledgeable expert in
    neurodegenerative diseases, particularly Alzheimer's disease.
Simultaneously, you are a helpful assistant who outputs responses
    in JSON format.
Return a JSON with the following schema: {schema}. You must
    strictly conform to this schema.'''

USER_PROMPT = f'''The pattern of neurodegenerative diseases in
    human brains is likely related to the joint effects of
    different brain graphs, where each graph node represents a
    region. Various brain connectivity or similarity matrices
    among brain regions can be used to represent these brain
    graphs.
I'm interested in the inference of a mixture of different graphs,
     which may serve as the substrate for disease appearance and
    progression in the brain, particularly for tau pathology in
    Alzheimer's disease.
Specifically, I have a list of {str(len(regions))} brain regions
    segmented using the Desikan-Killiany Atlas via FreeSurfer,
    namely {str(regions)}.
Now, for a specific region {ROI}, can you suggest which regions
    are related to {ROI} regarding the pathology appearance and
    progression of regional tau in human brains?
```

```
      Note that ctx_rh and ctx_lh in the region names are abbreviations
          for the cortex regions in the right and left hemispheres,
          respectively.
      Let's think step by step, considering each of the following
          factors:

      1. Similarity of cortical morphology
      2. Structural connectivity
      3. Microstructural profile covariance
      4. Spatial proximity
      5. Functional connectivity
      6. Any other possible patterns that can drive or affect the
          disease

      Please ignore the negative connections in the matrices mentioned
          above. Be open to more possible ways of connections. Less
          likely connections can be included but should be given a low
          strength level in the strength_dict.
      The output should be in JSON format with two keys, namely
          strength_dict and reasons.

      The strength_dict should be a dictionary whose keys are all the {
          str(len(regions))} region names from the provided region list
          , and the values should reflect the connection strength. The
          strength value should range between 0 and 1, with the
          following scale: 0-0.2 indicates very weak strength, 0.2-0.4
          indicates weak strength, 0.4-0.6 indicates moderate strength,
           0.6-0.8 indicates strong strength, and 0.8-1 indicates very
          strong strength. If there is no connection, the strength
          should be 0.
      The reasons should contain the corresponding explanations for the
           values in the strength_dict.'''
```

Listing 2: Prompt for synthetic data query of air pollution among main Chinese cities

```
      SYSTEM_PROMPT = f'''You are a knowledgeable expert in science
          such as physics, geography and neurosicence.
      Simultaneously, you are a helpful assistant who outputs responses
           in json format.
      Return a json with the following schema:{schema}. You must
          conform to the schema.'''

      USER_PROMPT = f'''The pattern of air pollution spreading across
          Chinese cities is likely related to the geodesic proximity
          among these cities, which can be represented by a graph where
           each graph node is a city.
      I am interested in inferring such a geodesic proximity graph that
           can serve as the foundation for understanding the appearance
           and progression of air pollution across different regions.
          Specifically, I have a list of {str(len(regions))} cities in
          China, namely {str(regions)}, where pollution is influenced
          by geographical proximity and prevailing wind patterns.
      Now, for a specific city {ROI}, could you suggest which cities
          are related to {ROI} with regard to the appearance and
          progression of air pollution based on their geodesic
          proximity?

      The output should be in JSON format with two keys, namely
          strength_dict and reasons.
      The strength_dict should be a dictionary whose keys are all the {
          str(len(regions))} region names in the provided region list
          and the values of the dictionary should reflect the strength
          of a connection.
```

```
        The strength value should be between 0 and 1, where the higher
            the value, the higher the strength of the connection. 0-0.2
            means weak connection, 0.2-0.4 means moderate connection, 0.4
            -0.6 means strong connection, 0.6-0.8 means very strong
            connection, and 0.8-1 means extremely strong connection.If
            there's no connection, the strength should be 0.
        The reasons should contain the corresponding reasons for the
            values in strength_dict.'''
```

## A.8 RELATED WORK

### A.8.1 DISEASE PROGRESSION MODELLING

Understanding the long-term trajectory of disease progression is crucial for advancing biological understanding, disease prevention strategies, and intervention development. Ideally, this goal would be achieved through densely sampled longitudinal measurements across an entire lifespan cohort. However, such an approach is often impractical due to patient inconvenience, cost considerations, and potential harm from repeated measurements. Additionally, early disease stages may lack characteristic symptoms, further hindering continuous monitoring from the very beginning. Disease Progression models are thus proposed to tackle these problems. These models can be broadly categorized into phenomenological and pathophysiological models Young et al. (2024a). Disease progression models usually estimate long-term disease trajectories alongside the corresponding temporal axis using cross-sectional data Fonteijn; Young (a); Firth; Young (b); Huang & Alexander; Venkatraghavan et al.; Tandon et al.; Parker et al.; Du & Zhou or irregularly sampled short-term data Severson (a;b); Villemagne; Samtani; Oxtoby to tackle the above-mentioned problems. In contrast, pathophysiological models, also known as mechanistic models, Seguin et al. (2023a); Vogel et al. (2023) incorporate the underlying pathological mechanisms to form disease trajectories. These include a variety of model types such as pathology appearance models, network models, and dynamic systems models. By leveraging both data and biological knowledge, pathophysiological models provide valuable insights for clinical applications.

Phenomenological models usually estimate disease progression trajectories alongside the corresponding temporal axis using cross-sectional data Fonteijn; Young (a); Firth; Young (b); Huang & Alexander; Venkatraghavan et al.; Tandon et al.; Parker et al.; Du & Zhou or irregularly sampled short-term data Severson (a;b); Villemagne; Samtani; ?; Oxtoby. In contrast, pathophysiological models, also known as mechanistic models, Seguin et al. (2023a); Vogel et al. (2023) incorporate the underlying pathological mechanisms to form disease trajectories. These include a variety of model types such as pathology appearance models, network models, and dynamic systems models. By leveraging both data and biological knowledge, pathophysiological models provide valuable insights for clinical applications.

### A.8.2 LLM FOR GRAPH LEARNING

Graph learning from time series, the task of uncovering the underlying variable-dependent relationships within a system, plays a critical role in various scientific fields Peters et al. (2017); Glymour et al. (2019). Causal graph discovery can be one typical representative of uncovering how the variables interact with each other. It often focuses on constructing DAGs, where edges represent causal influences between variables. However, a significant challenge in graph learning lies in identifying the unique true variable-dependent structure. Multiple DAGs can explain the observed data equally well, leading to the issue of non-identifiability Pearl (2009). While advancements such as restricting the data-generating process or employing deep learning for modelling variable covariances have been made, pinpointing the single correct graph solely from observational data remains an unsolved problem in many scenarios Kıcıman et al. (2023).

LLMs offer a promising perspective for addressing the challenges of graph learning by focusing on metadata associated with variables rather than their raw data values. By utilizing the contextual information embedded in variable names and problem domains, LLMs can infer graphs like human domain experts, based on general and domain-specific knowledge. Studies have explored the potential of LLMs for graph learning. Choi et al. Choi et al. (2022) demonstrated that LLM-generated prior hypotheses can enhance the accuracy of data-driven graph learning algorithms. Long et al. Long

et al. (2023) focused on LLMs as a post-processing step, showing their ability to reduce the size of a Markov equivalence class under the assumption of an optimal discovery algorithm output. Abdulaal et al. Abdulaal et al. (2023) proposed the CMA framework, which synergizes the metadata-based reasoning capabilities of LLMs with the data-driven modeling of DSCMs for graph learning.

### A.8.3 GRAPH LEARNING FOR TIME SERIES

Graph learning approaches in multivariate time series aim to uncover the causal relationship between time series. Such methods fall into several categories, including well-established approaches like Granger causality, alongside newer methods like constraint-based, score-based, and functional causal model-based approachesGong et al. (2023); Assaad et al. (2022).

Granger causality is one of the oldest tools for analyzing time series data and inferring potential variable-dependent relationships Granger (1969), forming the foundation for many modern methods. Earlier methods typically use the popular vector autoregressive (VAR) model under the assumption of linear time-series dynamics. However, real-world scenarios often involve non-linear dynamics, particularly in fields like neuroscience or finance Shojaie & Fox (2022). To address nonlinear dependencies, model-free methods like transfer entropy Vicente et al. (2011) and directed information Amblard & Michel (2011) offer an alternative, but they often require substantial data and struggle with high-dimensional settings. Beyond traditional and model-free approaches, researchers have explored other techniques to capture non-linear relationships in time series data. Differential equations excel at capturing non-linear relationships, making them valuable for describing interactions in dynamic systems. Recent work proposes Neural Graphical Models (NGMs), which model the latent vector field explicitly with penalized extensions to Neural ODEs Bellot et al. (2021). Other neural networks like MLP, RNN, LSTM can also be combined with Granger causality methods for modelling the complex and non-linear dynamics Gong et al. (2023); Shojaie & Fox (2022).

Another powerful tool for uncovering variable dependent relationships, constraint-based discovery methods work in two stages. First, it uses statistical tests to identify potential connections between variables, building a network of possible links. Then, specific rules are applied to orient these connections, resulting in a directed acyclic graph (DAG) that reflects the most basic causal structure between the variables. These approaches often rely on assumptions like the causal Markov property and faithfulness Gong et al. (2023). A prominent example is the Peter-Clark (PC) algorithm, which streamlines the process by reducing unnecessary tests, specifically for non-temporal data with the assumption of causal sufficiency. To handle time series data, the PC algorithm has been extended with methods like optimal causation entropy (oCSE) Sun et al. (2015), which leverages transfer entropy, and PCMCI Runge (2020) which uses momentary conditional independence tests.

Functional Causal Models (FCMs), also known as Structural Equation Models (SEMs) Neuberg (2003), describe a causal system using a set of equations. Each equation explains how a variable depends on its direct causes and an error term. This allows FCMs to capture both linear and non-linear relationships between variables. VAR-LiNGAM Hyvärinen et al. (2008; 2010), a typical FCM-based graph learning algorithm for time series, is built upon the non-temporal LiNGAM model Shimizu et al. (2006) and estimates structural autoregressive (SVAR) models by exploiting non-Gaussianity properties in the data. Another family of FCMs is based on the additive noise model (ANM), offering more flexibility by incorporating non-linear functions within its framework. It relaxes the linear constraints of VAR-LiNGAM and is suitable for more complex scenarios. An example of this family is the Time Series Models with Independent Noise (TiMINo) method Peters et al. (2013).

In score-based approaches, a graph corresponds to a probabilistic (or Bayesian) network; furthermore, a dynamic probabilistic (or dynamic Bayesian) network (DPN) is a probabilistic network in which variables are time series Assaad et al. (2022). score-based methods aim at finding sparse structural equation models that best explain the data, without any guarantee on the corresponding DAG (Kaiser and Sipos, 2021). This contrasts with, e.g., constraint-based approaches.

Score-based graph learning methods view variable-dependent relationships as a Bayesian network or a dynamic Bayesian network dealing with temporal data Assaad et al. (2022). Score-based methods prioritize finding a simple model that best explains the data, even if it does not perfectly map out the exact causal structure (DAG). This is in contrast to constraint-based methods, which focus on precisely identifying those causal connections. Friedman et al. Friedman et al. (2013) first use the Structural Expectation-Maximization (Structural EM) algorithm Friedman et al. (1997); FRIEDMAN

(1998) to infer a Dynamic Bayesian Network (DBN) from longitudinal data. Pamfil et al. Pamfil et al. (2020) proposed DYNOTEARS, a method that can simultaneously capture contemporaneous and time-lagged relationships between time series. To overcome the limitation of DYNOTEARS, which is a linear autoregressive model, NTS-NOTEARS Sun et al. (2021) is proposed based on 1D CNNs to extract both linear and non-linear relations among variables.

