# OpenReview forum: "LLM-guided spatio-temporal disease progression modelling"
_ICLR.cc/2025/Conference — Submitted to ICLR 2025_

### Official Review · Reviewer_Pz9o · 2024-10-31

**Soundness:** 3
**Presentation:** 3
**Contribution:** 3
**Rating:** 6
**Confidence:** 3

**Summary:**

The authors propose a novel LLM-guided spatio-temporal framework for modeling regional variable changes during disease progression. The key innovation of this paper lies in utilizing a large language model (LLM) to enhance graph initialization and regularization in both mechanistic and data-driven settings, demonstrating improvements in accuracy and convergence. The models were evaluated on regional tau-pathology propagation across the brain.

**Strengths:**

1. The paper is well-written and engaging. The idea of leveraging LLM knowledge for graph initialization is particularly intriguing. Given the recent advancements in LLM, effectively incorporating this knowledge into fundamental clinical research is crucial, and the authors demonstrate this integration in a good way.
2. The authors provide solid background information that helps readers easily grasp the context of their work.
3. Sufficient detail is offered on critical components, including prompting strategies, clear mathematical definitions, and experimental details, allowing readers to understand the underlying processes.
4. The evaluation of the model through critical edge numbers is both interesting and effective, showcasing the advantages of LLM-based graphs.
5. The authors conduct an extensive analysis, covering prompt ablation, synthetic data, different contexts (both data-driven and mechanistic), and model convergence.

**Weaknesses:**

1. I may have missed it, but the calculation of the metric "SSE" lacks a clear definition, making it challenging to assess its scale effectively.
2. The experiments appear to be conducted on a single data split, without cross-validation or random splitting. Given the relatively small variances reported in Table 2, it’s difficult to evaluate the significance of differences and reproducibility. Incorporating additional splits and reporting standard deviations would enhance the findings.
3. The representation of faster convergence (A.3), particularly between the second and third figures, could be clearer. Adjusting the scale might help elucidate this aspect.
4. While the authors emphasize the importance of identifiability and interpretability, their discussion primarily revolves around reducing the number of edges in the graph. There is limited exploration or discussions on how the derived graphs enhance understanding of regional interactions and differences during disease progression compared to other graphs.
5. In figure 4, descreasing the critical edge number throgu “7-factor” prompt is not very clear to me. It seems that the other factors provided by LLM does not lead to much advantages.
6. It appears that the model assumes a single typical progression path of phenotypes per disease, potentially overlooking disease heterogeneity. Recent research has demonstrated significant heterogeneity in disease progression, particularly in brain diseases, including amyloid and tau spreads.

**Questions:**

1. In Section A.3, why is the summation limited to four types of brain connectivities rather than five?
2. For clarification, when comparing Model 1 and Model 2, are the authors directly comparing the performance of these two methods, or are they comparing the graphs used in these models within the context of Method 3?

---

> ### Author Response · Authors · 2024-11-22
> **Response to Reviewer Pz9o**
>
> We thank the reviewer for carefully reviewing our work and providing detailed and insightful advice. Our responses are below.
>
> **1. Metric Definitions and Experimental Design**
>
> **Concern: Lack of cross-validation and unclear SSE definition.**
> **Response:**
>
> - Definitions for SSE (sum of squared error) and other metrics are provided in Section 2.3.
> - An updated 3-fold cross-validation results are now included (Tables 1 and 2).  For Table 2, the NGM experiments, the difference of sum squared error looks small since such neural ODE-based models have more trajectory parameters than mechanistic models in Table 1. Thus, in Table 2, our emphasis is more on the significantly smaller number of graph edges needed for better prediction offered by our proposed method, compared with the initial NGM, together with the more stable graph inference as shown in the newly added Figure 3 in the updated manuscript and the faster and more stable convergence shown in the Appendix A.2. Here better identifiability is offered from our method since the graph is biologically constrained with reasoning from LLM, rather than pure data-driven using the initial NGM method, where the obtained graph is very unstable, and much more graph edges are needed to maintain the performance, which has a risk of overfitting as well.
>
> **2. Concern: Assumption of a single disease progression path may overlook heterogeneity.**
>
> **Response:**
> Our model demonstrates proof of concept for simplified progression paths. Future work will explore subtyping to capture disease heterogeneity.
>
> **3. The unclear 7-factor prompt performance:**
> We have extended Figure 5 to show performance when the number of edges < 300, demonstrating how the 7-factor prompt allows maintenance of performance with fewer edges.
>
> **Responses to the questions:**
> - For the question regarding the section A.3 brain connectivities,
> The plot shows that several significant patterns of the LLM patterns can be found in four existing biological graphs, e.g. the top-left and bottom-right of the LLM graph is similar to the strongest signal in the structure and geodesic graph, while the sub-diagonal in the top-right and bottom-left of the LLM graph is similar to those in the morphological and microstructural graphs. We have now updated the plot with more graphs and deeper biological analysis to ensure that the graph from LLM corresponds to real biological graphs rather than hallucination.
>
> - For clarification of model comparison,
> In real data scenarios, we evaluate the models using the tau prediction accuracy as a downstream task with different embedded graphs due to the lack of ground truth graphs in AD. But in synthetic data experiments in Appendix A.1 we have been able to compare more baseline models directly on the graph with ground truth.

---

### Official Review · Reviewer_YR4J · 2024-11-01

**Soundness:** 2
**Presentation:** 2
**Contribution:** 2
**Rating:** 5
**Confidence:** 4

**Summary:**

This paper introduces a novel spatio-temporal framework guided by large language models (LLMs) to model long-term disease progression on brain graphs, utilizing structure discovery from longitudinal, population-level patient data. A key innovation lies in addressing the unknown temporal alignment of observations by embedding the method within a dual-optimization framework. This framework simultaneously estimates the disease progression model—capturing common biomarker trajectories across a population—and performs structure discovery, learning a graph that represents the spatiotemporal interactions among biomarkers. The proposed approach achieves superior prediction accuracy, faster and more stable convergence, and enhances the interpretability of the model.

**Strengths:**

This paper explores the use of Large Language Models (LLMs) as expert guides to enhance graph inference by integrating expert knowledge.

**Weaknesses:**

1. There are a lot of typos, such as "To address these limitations, we **consder** use Large Language Models (LLMs) as
expert guides to enforce the graph inference with expert **knowlegdge**" and so on.
2. In Eq. (5), why **p**  is calculated from the **99th** percentile of the tau distribution at each region?
3. In Eq. (8), what does $D_{LLM}$ represent? Could you clarify the statement, "This not only accounts for the diffusion process along the white matter bundles but also considers additional factors related to tau accumulation from the knowledge base of various LLMs"? From Equations (7-9), I am not seeing how structural connectivity is addressed (i.e., the diffusion process along the white matter bundles).
4. In Eq. (10),  what does $X$ stand for? what's the relationship between this section and the previous sections? I can't follow this.
5. What's the dataset you use? How about the performance on other diseases? What's the sample size? How many time point for each subject do you use?
6. Insufficient experiments and lack of comparison of many disease progression modeling models.
7. This method claims to estimate regional tau-pathology propagation over the brain. Can authors provide the related propagation pattern in the brain over time?

**Questions:**

Please refer to weaknesses.

---

> ### Author Response · Authors · 2024-11-22
> **Response to Reviewer YR4J**
>
> We thank the reviewers for the reviews. Please find below our responses.
>
> **1. Typos and Notational Issues**
>
> **Concern: Typos and unclear notation in equations.**
>
> **Response:**
>
> - All typographical errors have been corrected.
> - Explanations between paragraphs have been added for consistency.
> - Definitions of unclear notations have been added.
> Following established work (Chaggar et al.), the 99th percentile is used for tau distribution to represent region-specific carrying capacity. This approach recognizes that different brain regions reach different maximum tau concentrations at the disease end-stage (thus using a very high percentile) rather than assuming uniform levels.
> - For equation 8, $\mathbf{D_{LLM}}$ represents the diagonal degree matrix used in graph Laplacian calculations, consistent with the notation in equation 5, which has been defined.
> - We have added a more detailed explanation of how structural connectivity and other factors are addressed in the framework of model 3 in the new manuscript by including the command for the LLM to explicitly consider these connectome factors in the prompt (defined in Appendix A.7). Appendix A.3 further verifies that the LLM graph corresponds to real biological graph patterns.
> We have added the definition of X to equation 10 now in the paper. Regarding the relationship between this section (data-driven neural graphic modelling) and the previous sections (mechanistic modelling), we have demonstrated our method in both the mechanistic model and the data-driven model. Equation 10 describes the method of data-driven graph learning, with results corresponding to Table 2 and Section 3.3.
>
> **2. Small Dataset**
> Concern: Lack of dataset description:
> Response: We have mentioned the detailed dataset description in Appendix A.4.  We use the ADNI (Alzheimer's Disease Neuroimaging Initiative) dataset.
>
> **3. Other concerns**
>
> - **Performance on Other Diseases:**
> This work serves as a proof of concept demonstrating how LLMs can enhance graph construction for mathematical models of disease spreading. The framework could potentially be adapted for other diseases.
>
> - **Lack of comparisons with other disease progression models:**
>
> Our framework can enhance graph inference for mechanistic disease progression models. While we demonstrate this using network diffusion models, the approach could be applied to any mechanistic model with an embedded graph in future work. The comparison models we selected are representative of disease progression models. Our goal is to demonstrate how incorporating LLMs can enhance mechanistic models rather than conducting an exhaustive comparison with all possible mechanistic models and data-driven models.
>
> - **Propagation Patterns:**
>
> We have added brain visualization plots showing tau propagation patterns over time in Appendix A.5.
>
> **References:**
> Chaggar, P., Vogel, J., Pichet Binette, A., Thompson, T. B., Strandberg, O., Mattsson-Carlgren, N., ... & Goriely, A. (2023). Personalised regional modelling predicts tau progression in the human brain. bioRxiv, 2023-09.

---

> > ### Comment · Reviewer_YR4J · 2024-11-23
> > **Thanks**
> >
> > Thank you for your rebuttal, I'd like to raise my score, but my primary concerns (small dataset and insufficient experimentation) have not been effectively addressed.

---

> ### Author Response · Authors · 2024-11-24
> **Further responses for the remaining two concerns**
>
> We thank the reviewer for the timely response and careful reconsideration. We want to address the two remaining concerns further.
>
> **Small dataset**
>
> Regarding the sample size, we acknowledge that our data size is relatively small. However, this is a well-recognized challenge in the field of longitudinal tau PET studies, stemming from multiple factors: high costs of PET scanning, radiation exposure concerns, and significant challenges in maintaining long-term follow-up visits. For context, recent comparable studies have used similar sample sizes - Leuzy et al. (JAMA Neurology 2023) included 215 subjects in their longitudinal tau analysis, while Yang et al. (Neuroimage 2022) analysed fewer than 100 subjects for tau prediction using a similar mechanical model along the structural connectome. Binette et al.
>
> In fact, a key motivation for developing the methods here is that data sets are small and will remain so for the foreseeable future. Here, we conduct experiments for mechanistic and data-driven neural graphical models. It is worth noticing that mechanistic models have relatively simple and explicit math equations (section 2.5.1) that only have 2 parameters $\alpha$ and $k$ that define the speed of tau diffusion and production, respectively, to define the slope of the trajectory for longitudinal progression, thus don't need that much longitudinal data. For the data-driven models with more complex parameterization (section 2.5.2), we demonstrate that, in this case, the initial regularization of NGM does not work well by offering unstable graph inference (Figure 3), which means that embedding biological knowledge into the model is essential to reduce parameter counts from fully data-driven models. Of course, this requires accurate biological modelling, and our aim here is to address a key inaccuracy in previous models – the brain connectome – by exploiting knowledge contained in modern LLMs.
>
> Furthermore, the small data size makes understanding the full disease progression process challenging using existing methods. Thus our framework was specifically designed to address the sparse, irregularly sampled, limited number of longitudinal collections of Tau. Our method innovatively leverages sparse, short-term longitudinal or cross-sectional individual-level data to construct long-term cohort-level disease progression trajectories to understand the disease progression trajectory from the very early to the very late disease stage. After the training, we will be able to allocate any new subject on the relative location of the trajectory to understand their staging and thus provide a better diagnosis.
>
> **Insufficient experiments**
>
> For the lack of comparison with other disease progression modelling, we emphasise that LLM can help provide a more accurate and robust graph when given a specific mechanistic model that needs an embedded graph (either given an existing connectome or inferring a graph from a driven method). Thus, it makes more sense to compare the model embedded with the initial graph to the one embedded with an LLM graph rather than compare the current model with other disease progression models. For instance, since disease progression models like EBM [3] and SuStaIn [4] are purely data-driven without any constraints from the graphs and with different parameterizations, direct comparisons are not meaningful.
>
> **References**
>
> [1] Leuzy, A., Binette, A. P., Vogel, J. W., Klein, G., Borroni, E., Tonietto, M., ... & Alzheimer’s Disease Neuroimaging Initiative. (2023). Comparison of group-level and individualized brain regions for measuring change in longitudinal tau positron emission tomography in Alzheimer disease. JAMA neurology, 80(6), 614-623.
>
> [2] Yang, F., Chowdhury, S. R., Jacobs, H. I., Sepulcre, J., Wedeen, V. J., Johnson, K. A., & Dutta, J. (2021). Longitudinal predictive modeling of tau progression along the structural connectome. Neuroimage, 237, 118126.
>
> [3] Fonteijn, H. M., Modat, M., Clarkson, M. J., Barnes, J., Lehmann, M., Hobbs, N. Z., ... & Alexander, D. C. (2012). An event-based model for disease progression and its application in familial Alzheimer's disease and Huntington's disease. NeuroImage, 60(3), 1880-1889.
>
> [4] Young, A. L., Marinescu, R. V., Oxtoby, N. P., Bocchetta, M., Yong, K., Firth, N. C., ... & Alexander, D. C. (2018). Uncovering the heterogeneity and temporal complexity of neurodegenerative diseases with Subtype and Stage Inference. Nature communications, 9(1), 4273.

---

> > ### Author Response · Authors · 2024-12-02
> > **Further reply to Reviewer YR4J**
> >
> > Dear Reviewer YR4J,
> >
> > As the rebuttal period is nearly over, we hope our follow-up responses have addressed your concerns and provided greater clarity regarding our work based on your insightful reviews.
> >
> > We greatly appreciate your time and effort in evaluating our submission. If there are any remaining questions or if further clarification would be helpful, please feel free to reach out to us before the discussion concludes.
> > Thank you once again for your valuable feedback and engagement.
> >
> > Best regards,
> > The Authors

---

### Official Review · Reviewer_Kk5d · 2024-11-03

**Soundness:** 1
**Presentation:** 1
**Contribution:** 1
**Rating:** 1
**Confidence:** 5

**Summary:**

This work presented a proof-of-concept approach to model Alzheimer's disease (AD) using LLM. The idea is interesting. However, the feasibility and power of LLM in understanding the etiology of AD is not clearly demonstrated. Our current understanding on disease mechanism is still elusive in the AD field. Not sure how the proposed approach could advance our current understanding on AD.

Another main concern is the data sample size. Neuroimaging data from ADNI is used in this work. There is only couple hundreds (<300) subjects having both modalities (structural connectome and pathology data). It is not clear how to address the issue small sample size.

Also, the data pre-processsing is not clear. Which brain atlas is used? Based on the node number, looks like the Desikan-Killiany Atlas. Please note this atlas does not include sub-cortical regions, means it is not a perfect brain parcellation for studying AD.

Result interpolation lacks biological underpinnings and supporting evidence from current literatures in AD.

**Strengths:**

First try of LLM in understanding disease progression.

**Weaknesses:**

Not clear how this approach distinct to other knowledge graph approaches.
Lack of a killer application supporting the motivation why we need invent the new wheel for AD.
Figures are not very helpful to understand the method.

**Questions:**

What is the essential motivation of employing LLM in modeling AD progression?

---

> ### Author Response · Authors · 2024-11-21
> **Response to Reviewer Kk5d**
>
> Please find our responses below.
>
> **1. LLM for Disease Progression**
>
> **Concern: Feasibility and Motivation for Using LLM in AD Modelling**
>
> **Response:**
>
> **Limitations of Current Approaches:**
>
> There are two ways to model the disease progression within the brain’s dynamical system, each with weaknesses:
> - Mechanistic models, such as network diffusion models, are widely used to understand disease mechanisms. These models have explicit diffusion equations, providing key insights to aid disease understanding and treatment development.  However, these models rely heavily on connectivity graphs, which are typically derived from structural connectomes. These imperfect connectomes often represent only linear combinations of limited biological modalities.
> - Data-driven models in the neural graphical model structure learn the graph and trajectory purely from data with regularization, but such graph learning lacks interpretability and identifiability (which demonstrates that when the dimension of the graph is high, the different data-driven graphs can offer similar results.)
>
> With the usage of LLM, we have demonstrated in this paper that the strengths from both sides of the models can be addressed:
> - **Enhanced Graph Inference for mechanistic model:**
> Our LLM-based approach integrates multiple data modalities, without relying on any database of brain connectomes or other biological information such as gene databases or neurotransmitter systems. The LLM can integrate information about different disease mechanisms into a unified framework. This enables more accurate representation of disease mechanisms.
> - **Technical Advantages for data-driven model:**
> LLM-based constraints ensure stable graph structures with fewer learnable parameters than data-driven approaches, while data-driven methods provide unstable graphs.  Our system shows faster convergence in tau fitting compared to the initial neural graphical models and higher accuracy in progression modelling.
> - **Improved Interpretability and Reasoning:**
> The LLM acts as an expert system that provides explicit reasoning for connections and explains biological mechanisms. Unlike traditional knowledge graphs, it offers transparent justification for each relationship in the network.
> - **Novel Insights to AD:**
> Abdulaal et al., (ICLR 2023) [1] shows LLMs can discover novel AD causal relationships missed by human experts while validating findings against existing literature.
> We demonstrated that synthetic experiments show better recovery of ground truth relationships Real-world validation confirms more accurate tau spread prediction and improved alignment with known disease mechanisms.
>
> **Distinctions from Traditional Knowledge Graphs**
> - Flexibility: Users can easily put their hypotheses of which mechanistic factors to include in the graph building by adjusting the prompt and testing the graph in the disease progression modelling instead of relying on any existing database. Adding a sentence like “please think about any further possible factors” can encourage LLM to put in its new insight. Where the knowledge graph has less flexibility.
> - Reasoning Capability: Beyond simple connections, our system provides mechanistic explanations for each relationship.
> - Adaptability: LLM rapidly incorporates new research and adjusts relationships based on evidence.
>
> In conclusion, LLM-based graphs offer advancements by integrating multimodal data, enhancing interpretability, and improving graph identifiability. They address the limitations of existing connectivity graph and provide novel insights into AD progression.
>
> **2. Small Sample Size**
>
> **Concern: Limited data from ADNI raises concerns about model generalizability.**
>
> **Response:**
> We acknowledge the challenge of limited longitudinal tau PET data due to high cost and challenges in carrying out follow-up visits, etc. Thus this is a standard data set size for this kind of work. For instance, Leuzy et al (JAMA Neurol  2023 )[2] uses 215 subjects in the longitudinal Tau study and Yang et al ( Neuroimage 2022)[3] uses less than 100 subjects for tau prediction using a similar mechanics model. That’s one of the reasons that we propose such a framework to leverage sparse, cross-sectional data to construct long-term trajectories to tackle such challenges. What’s more, we embed expert constraints from LLM for model sparsity, thus enhancing the robustness of learning in small data size scenarios. Future work will explore integrating larger datasets and additional biomarkers to validate scalability.
>
> **3. Brain Atlas and Subcortical Regions**
>
> **Concern: Use of Desikan-Killiany atlas without subcortical regions.**
>
> **Response:**
> Desikan-Killiany atlas was chosen due to its prevalence in AD studies, including the ADNI dataset, which is very commonly used in the field. Subcortical regions were excluded to avoid confounding due to off-target binding of the AV1451 tau PET tracer [4]. Relevant citations have been added to support this decision.

---

> > ### Comment · Reviewer_Kk5d · 2024-11-23
> >
> > Thanks for the rebuttal. The biological mechanism for AD is largely elusive. In addition, mounting evidence shows that AD is shaped by the complex interaction between genes and the environment. There is a surge of using LLM in the biomedical field, but mostly relevant to cancer and non-brain diseases. Overall, there is less evidence to support the rigor of this approach. I appreciate the merits of the technique in this work. However, a lack of scientific evidence reduces enthusiasm for this work.

---

> ### Author Response · Authors · 2024-11-21
> **References for Reviewer Kk5d**
>
> **References:**
>
> [1] Abdulaal, A., Montana-Brown, N., He, T., Ijishakin, A., Drobnjak, I., Castro, D. C., & Alexander, D. C. (2023). Causal Modelling Agents: Causal Graph Discovery through Synergising Metadata-and Data-driven Reasoning. In The Twelfth International Conference on Learning Representations.
>
> [2] Leuzy, A., Binette, A. P., Vogel, J. W., Klein, G., Borroni, E., Tonietto, M., ... & Alzheimer’s Disease Neuroimaging Initiative. (2023). Comparison of group-level and individualized brain regions for measuring change in longitudinal tau positron emission tomography in Alzheimer disease. JAMA neurology, 80(6), 614-623.
>
> [3] Yang, F., Chowdhury, S. R., Jacobs, H. I., Sepulcre, J., Wedeen, V. J., Johnson, K. A., & Dutta, J. (2021). Longitudinal predictive modeling of tau progression along the structural connectome. Neuroimage, 237, 118126.
>
> [4] Ricci, M., Cimini, A., Camedda, R., Chiaravalloti, A., & Schillaci, O. (2021). Tau biomarkers in dementia: positron emission tomography radiopharmaceuticals in tauopathy assessment and future perspective. International Journal of Molecular Sciences, 22(23), 13002.

---

> ### Author Response · Authors · 2024-11-24
> **Further reponses to Reviewer Kk5d about scientific evidence of LLM for brain research**
>
> We appreciate the reply. However, we would like to emphasise that we do not attempt or claim to produce a comprehensive model of the mechanisms of AD here. We fully agree with the reviewer that major challenges still remain in that endeavour. Our work simply aims to demonstrate how judicious use of LLMs can improve the basic components of such models, specifically estimated brain connectomes, and bring us closer to providing insight into disease mechanisms via computational modelling.  This addresses a major need in computational medicine.  For example, a recent review (Vogel et al. Nature Reviews Neuroscience 2023) [1] specifically highlights the lack of accuracy in connectomes as a key issue with mechanistic spreading models. We provide the first experiments that demonstrate amelioration via LLM usage.
>
> The application of Large Language Models (LLMs) in brain research is gaining significant scientific validation, as evidenced by recent publications in prestigious journals such as Neuron. Notably, Bzdok et al. (2024)[2] present a comprehensive perspective on how LLMs can revolutionize neuroscience research, with particular relevance to brain connectome analysis. There are some key factors demonstrated in their article:
> - LLMs excel at integrating multi-modal brain data and semantic content, which is crucial for our work on brain connectome inference using the combination of multi-modal factors. Their ability to process and synthesize complex brain-imaging data alongside other data modalities enables a more comprehensive understanding of neural interactions and disease progression patterns.
> - LLMs are particularly powerful in analyzing brain-behavior relationships and network dynamics. This capability directly supports our approach to modelling how different brain regions interact with each other, as LLMs can help identify and characterize complex patterns in neural connectivity that might be missed by traditional analytical methods.
> - LLMs' demonstrated ability to break down silos between neuroscience sub-disciplines is especially valuable for connectome research. By synthesizing knowledge across different domains of neuroscience from molecular to behavioural levels,  LLMs can provide unique insights into how brain networks function and evolve during disease progression. This cross-disciplinary integration is essential for developing more accurate and comprehensive models of brain connectivity.
> These capabilities make LLMs particularly well-suited for advancing our understanding of brain connectomes and their role in disease progression, representing a rigorous and promising approach in modern neuroscience research.
>
> **References**
>
> [1] Vogel, J. W., Corriveau-Lecavalier, N., Franzmeier, N., Pereira, J. B., Brown, J. A., Maass, A., ... & Ewers, M. (2023). Connectome-based modelling of neurodegenerative diseases: towards precision medicine and mechanistic insight. Nature Reviews Neuroscience, 24(10), 620-639.
>
> [2] Bzdok, D., Thieme, A., Levkovskyy, O., Wren, P., Ray, T., & Reddy, S. (2024). Data science opportunities of large language models for neuroscience and biomedicine. Neuron, 112(5), 698-717.

---

### Official Review · Reviewer_erUT · 2024-11-04

**Soundness:** 2
**Presentation:** 2
**Contribution:** 2
**Rating:** 3
**Confidence:** 5

**Summary:**

This paper presents a novel disease progression model for neurodegenerative diseases that leverages a mixture of large language models (LLMs) to infer brain region interactions for predicting temporal dynamics. The approach uses prompts to query LLMs about potential interactions between brain regions, which are then used to construct a graph that informs disease progression

**Strengths:**

### Strengths

1. **Innovative Use of LLMs in Disease Progression Modeling (DPM)**:
The paper introduces a creative and potentially impactful approach by leveraging large language models (LLMs) for graph-based disease progression modeling  in neurodegenerative diseases.

**Weaknesses:**

### Weaknesses

1. **Vague Language and Disorganized Manuscript**:
   The paper uses terms like "causal interactions" without justification. The use of graph learning alone does not establish causation, as this typically requires specific assumptions or experimental design elements, such as interventions or causal inference techniques. Without these, the inferred relationships are associative rather than causal. Also,  the term "long-term" is not clearly defined, despite its critical importance in temporal modeling of disease progression. The lack of a precise timeframe (e.g., years or decades) makes it difficult to understand the model's intended scope and applicability. A clearer definition of "long-term" would strengthen the paper. Additionally,  there are instances where equations or concepts are referenced before they are defined, disrupting the logical flow and making it difficult for readers to follow. For instance, equation 9 is mentioned in the text (Section 2.2) before it is officially introduced (Section 2.5.1). Finally, there are several typos throughout the manuscript. Minor notice, but important, evaluation measures like SSE should be introduced first and then utilized with abbreviation for more clarity.

2. **Lack of Comparisons with Baselines and Limited Qualitative Results**:
   Although the authors cite several relevant baseline methods, such as Bellot et al. (2021), they do not provide empirical comparisons. Without benchmarking against these established approaches, it is difficult to assess the advantages of the proposed method.
Also, there are no qualitative results with predicted trajectories that would showcase that the method works in several test examples.
Additionally, in section 3.3 and 3.4 of the results authors use only the Claude3.5 LLM. However, their proposed method is the mixture of the LLMs and not only a single LLM (Claude3.5). This inconsistency, weakens result's coherence  as well as the validity of the proposed method.

3. **Questionable Prompt Strategy for LLMs**:
   The approach relies on querying LLMs with prompts for each brain region to infer relationships, which raises several concerns:
   - **Scientific Validity**: General-purpose LLMs lack domain-specific scientific knowledge and are prone to hallucinations. This means that the generated weights or connections could be inaccurate or even fabricated.
   - **Scalability**: The strategy of creating prompts for each brain region does not scale well for fine grained ROI brain representations, making it computationally intensive and potentially inconsistent.
   - **Stability and Consistency**: Since LLM responses can vary with prompt phrasing, it’s unclear if these outputs are reliable across prompts. An analysis of prompt sensitivity or consistency checks would be necessary to validate this approach.

4. **Limited Longitudinal Data for Temporal Modeling**:
   With only 254 samples from 185 subjects, it appears that most subjects have only a single acquisition (baseline) and one follow-up. This data limitation raises doubts about the model’s ability to learn meaningful temporal dynamics. The authors do not address how the model captures progression trends with such limited longitudinal data, leaving the temporal component of the model questionable.

5. **Limited Discussion of Practical Application and Real-World Feasibility**:
    The paper lacks a detailed discussion of how the proposed Disease Progression Model could be applied in real clinical or research settings. The reliance on prompts and general-purpose LLMs may hinder broader adoption, as access to specific LLMs, prompt engineering, and technical resources may not be feasible in clinical environments. Addressing these practical challenges and offering guidance for real-world application would enhance the paper's relevance.

**Questions:**

### Questions

1. **Handling of Repeated Measures**:
   - "Authors mention that they leverage longitudinal population-data. How do you handle repeated measures in the training process?"

2. **Ratio Specification for Mixing LLMs**:
   - "How do you specify the ratio (0.865:0.135) for mixing the two LLMs? Could you provide a rationale or any empirical basis for choosing this specific ratio?"

3. **Cross-Validation for Robustness**:
   - "A simple splitting of the 255 subjects into train-test-validation is not enough. A fold cross-validation (train-test-validation) splitting multiple times would produce more powerful results. Could you comment on the reason for choosing a single split rather than cross-validation?"

---

> ### Author Response · Authors · 2024-11-21
> **Response to Reviewer erUT**
>
> We appreciate the constructive comments from the reviewer. It seems that several raised issues, like the lack of comparison with baseline models (Bellot etc.), the lack of definition of the LLM mixture ratios, and the lack of validation of the LLM graphs, were already included in the initial paper submission. However, we have reorganised the manuscript and adjusted the text to improve clarity, so that these experiments are easier to find. We address these and other issues in more detail below.
>
> **1. Vague Language and Disorganized Manuscript**
>
> **Concern: Use of terms like "causal interactions" and lack of definition for "long-term."**
>
> **Response:**
>
> - We clarified that "causal" refers to "mechanistic interactions" and replaced it.
> - "Long-term progression" is now defined as cohort-level disease progression from early onset to late stage. The time axis represents relative disease progression (pseudo-time), with preserved actual time gaps for longitudinal data from the same subject.
> - We reorganised the manuscript to introduce concepts and equations logically and added clear definitions for all evaluation metrics before their use.
>
> **2. Lack of Baseline Comparisons and Limited Qualitative Results**
>
> **Concern: No baseline comparisons with models like Bellot et al. (2021) and lack of qualitative test results.**
>
> **Response:**
>
> - Table 2 already compared our method's regularisation with Bellot et al.'s Neural Graphical Model (NGM), showing faster convergence, better accuracy, and significantly fewer parameters in sparser graphs, indicating robust learning and reduced overfitting.
> - Synthetic data experiments (Appendix A.1) include comparisons with other graph-learning methods (DCM, PCMCI, SWAM), where the learned graphs are compared with the ground truth.
> - Tables 1 and 2 include quantitative test results of metrics like SSE and AIC, defined and updated with 3-fold cross-validation. The framework generates long-term disease trajectories from short-term snapshots, allocating test set subjects using the "time optimization step."
>
> **3. Inconsistency of Usage of LLM Between Sections 3.3 & 3.4 and Previous Sections**
>
> **Concern: The proposed method involves a mixture of LLMs, but sections 3.3 and 3.4 show only Claude 3.5 results.**
>
> **Response:**
>
> - Sections 3.3 and 3.4 used Claude 3.5 as a proof-of-concept to analyze LLM-specific performance related to prompts and reasoning. - Appendix A.6 now include reasoning examples from other LLMs. Table 2 includes performance data for the LLM-mixture graph in the NGM modelling task.
>
> **4. Prompt Validity and Scalability**
>
> **Concern: LLM-generated prompts may lack scientific validity and scalability.**
>
> **Response:**
>
> - Scientific Validity: Appendix A.1 shows that LLM-generated graphs align closely with ground truth in synthetic data experiments, and Appendix A.3 (Figure 8) shows patterns consistent with real biological graphs.
> - Scalability: Queries balance accuracy and computational load by querying the interaction of one region to the rest regions as a 68-length array (68 queries total). Pairwise querying to form an element of a matrix ($68\times68$ queries needed) is more accurate but computationally expensive, while full $68\times68$ matrix queries all at once will cause reasoning challenges. Parallel querying ensures scalability and efficiency.
> - Consistency: A 0-1 strength scale (Appendix A.7) defines levels of strength for consistent querying.
>
> **5. Small Sample Size**
>
> **Concern: Limited data from ADNI raises concerns about model generalizability.**
>
> **Response:**
>
> Tau PET datasets are inherently small due to imaging costs and follow-up challenges. For example, Leuzy et al. (2023) studied 215 subjects, and Yang et al. (2022) used fewer than 100 in similar models. Our framework addresses this limitation by reconstructing long-term trajectories from sparse cross-sectional data, leveraging LLM-derived constraints for sparsity and robustness. Future work will expand datasets and biomarkers to evaluate scalability.
>
> **Questions Raised:**
> - Handling Repeated Measures: Longitudinal scans preserve true time gaps, improving temporal consistency. Despite the small sample size, the temporal axis is referred to as "relative time," focusing on relative subject positions along the trajectory.
> - The LLM mixture ratio (0.865:0.135) was described in Section 2.4.3 of the initial submission, with hyperparameter $\alpha$ tuned on the validation set.
> - Tables 1 and 2 now reflect 3-fold cross-validation results.
> - Future work highlights potential applications of the framework in other fields leveraging LLM-derived knowledge.
>
> **References**
>
> [1] Leuzy, A. et a; (2023). Comparison of group-level and individualized brain regions for measuring change in longitudinal tau positron emission tomography in Alzheimer disease. JAMA neurology, 80(6), 614-623.
>
> [2] Yang, F. et al (2021). Longitudinal predictive modeling of tau progression along the structural connectome. Neuroimage, 237, 118126.

---

> > ### Comment · Reviewer_erUT · 2024-11-22
> > **Response**
> >
> > Thank you for the effort you put into addressing my comments, providing clarifications, and making several corrections.
> >
> > However, I still find that this work has significant limitations. The qualitative examples that I requested also are pretty vague and not explained at all. For example what the colorbar visualizes? Most importantly, I remain unconvinced about the practical applicability of this method in clinical practice.
> >
> > For that I will maintain my score since I believe that this line work is not ideal for publication.

---

> ### Author Response · Authors · 2024-11-27
> **Further reply to Reviewer erUT: Clinical application of the proposed method**
>
> We appreciate your feedback. For the visualization of tau distribution via brain mapping with the disease progression (Appendix A.5 Figure 10), we've added a more detailed description, such as the colour bar displaying the level of tau concentration at each brain region. We can explain any other aspects that you find vague.
>
> Additionally, we want to clarify the clinical applications of our proposed framework.
>
> Disease progression models, in general, provide staging systems that enable us to place individual patients along a cohort-level trajectory. In the updated manuscript, we have added Figure 9 in Appendix A.5 to illustrate this process. This figure shows how the trajectory is constructed using snapshots of the individual training subjects. Then, when a new subject arrives, it can be placed appropriately on the trajectory (indicated in orange). This placement determines the disease stage of the subject relative to the entire cohort, which is critical for diagnosis. Importantly, the disease stage reflects the extent of pathology progression, which can inform treatment decisions and care planning.
>
> Besides, by integrating graph inference with LLM and obtaining a better graph as the substrate of mechanistic disease progression, we can gain deeper insights into the mechanisms of Alzheimer's disease (AD), such as how different brain regions interact with each other, as displayed by the inferred graph, together with the corresponding detailed reasoning from LLM. This approach can help identify key factors that drive tau spreading between specific brain regions, ultimately enabling more effective treatment strategies. This addresses a major need in computational medicine. Recent reviews, for example (Young et al. Nature Reviews Neuroscience 2024; Vogel et al. Nature Reviews Neuroscience 2023), specifically highlight the lack of accuracy in connectomes as a key issue with mechanistic spreading models.
>
> **References**
>
> [1] Young, A. L., Oxtoby, N. P., Garbarino, S., Fox, N. C., Barkhof, F., Schott, J. M., & Alexander, D. C. (2024). Data-driven modelling of neurodegenerative disease progression: thinking outside the black box. Nature Reviews Neuroscience, 25(2), 111-130.
>
> [2] Vogel, J. W., Corriveau-Lecavalier, N., Franzmeier, N., Pereira, J. B., Brown, J. A., Maass, A., ... & Ewers, M. (2023). Connectome-based modelling of neurodegenerative diseases: towards precision medicine and mechanistic insight. Nature Reviews Neuroscience, 24(10), 620-639.

---

> > ### Author Response · Authors · 2024-12-02
> > **Further reply to Reviewer erUT**
> >
> > Dear Reviewer erUT,
> >
> > As the rebuttal period is nearly over, we hope our follow-up responses have addressed your concerns and provided greater clarity regarding our work based on your insightful reviews.
> >
> > We greatly appreciate your time and effort in evaluating our submission. If there are any remaining questions or if further clarification would be helpful, please feel free to reach out to us before the discussion concludes. Thank you once again for your valuable feedback and engagement.
> >
> > Best regards,
> >
> > The authors

---

### Author Response · Authors · 2024-12-01
**Summarized responses to all the reviewers. Looking forward to further replies**

We thank all reviewers for their time, effort, and valuable feedback, and we are looking forward to the follow-up replies to our responses. We appreciate the recognition of the novelty and innovation of our approach, particularly the use of Large Language Models as an expert to enhance graph learning for disease progression modelling (erUT, YR4J, Pz9o). We also value the positive feedback on the clarity of our background information, clear mathematical framework, and comprehensive analysis, covering prompt ablation, synthetic data, different contexts (both data-driven and mechanistic), and model convergence (Pz9o).
In response to the constructive critiques, we have made several improvements to the manuscript, addressing concerns across clarity, methodology, and experimental rigour. Key updates include:

**1.	Small Dataset Concerns (Kk5d, YR4J, Pz9o):**
- Contextualized our sample size within field standards, citing comparable recent studies using similar or smaller cohorts
- Explained how our framework addresses sparse data challenges through:

   * Simple parameterisation in mechanistic models
   * LLM-derived constraints reducing parameter count
   * The construction of long-term trajectories from irregular, sparse and limited snapshots is the precise contribution of our proposed framework to address such a problem.

**2.	Experiments for Graph Validity and Model Comparison:**
 - Included 3-fold cross-validation results in Tables 1 and 2. (erUT, Pz9o).
- Clarified the experiments had been included in our initial submission, including the comparison with Bellot et al. (2021) in Table 2, validating the LLM graph by comparing it with the existing biological graphs (Appendix A.3) and other baseline models of graph inference for time series (Appendix A.1) (erUT).
- Added more results using the mixture of LLM in Table 2  (erUT).
- Emphasized how LLM constraints improve graph stability and reduce overfitting in mechanistic and data-driven models. Added Figure 3, showing that the initial methods without expertise provide different graph inference outcomes. (Kk5d, Pz9o).

**3.	Clinical Application:**
- Clarified the clinical applicability of the proposed framework, particularly in disease staging and mechanistic insights into tau propagation. Figures 9 and 10 were added to describe the disease staging and quantitative visualization of predicted disease patterns in Appendix A.5, with a discussion on practical applications in disease staging and diagnosis.
- Referenced recent reviews highlighting improving connectome accuracy and understanding as a key challenge and explained how we improve graph inference and interpretation using LLMs. (erUT, Kk5d).

**4.	Clarified the Motivation for using LLM (Kk5d)**
- Cited recent publications validating LLM applications in brain research and LLMs' unique capabilities in integrating multi-modal brain data and analysing brain-behaviour relationships and network dynamics.
- Explained how expertise from LLMs addresses limitations of both mechanistic models (combining multi-modal connectomes to obtain high accuracy and fewer learnable parameters) and data-driven approaches (enhancing both identifiability and interpretability with faster and smoother convergence)
- Emphasized that our goal is to improve the graph component in existing models, which has been a known limitation waiting to be solved, mentioned by several nature neuroscience review papers.
- Explained the advantages of LLM compared with traditional knowledge graph

**5.	Manuscript Clarity:**
- Enhanced logical flow and definitions in Tables 1 and 2 (erUT, YR4J).
- Added detailed explanations and corrected typos and unclear notation (YR4J, Pz9o).

We hope these enhancements address the reviewers’ feedback and further highlight the rigor and applicability of our work.  We thank the reviewers again for their constructive insights and the AC for considering our submission. Any further discussion from the reviewers would be highly appreciated.

---

> ### Comment · Reviewer_Kk5d · 2024-12-01
>
> In my opinion, this work is pre-mature and lacks a clear biological motivation. The clinic benefit is not clear either. Not sure LLM can generate new knowledge that is biologically meaningful. There is no metric to evaluate new "findings". Our current understanding on AD/ADRD is still very limited. With that being said, it might be too early to apply LLM. My suggestion is that LLM might be useful in some well-controlled studies (targeting specific cohorts) instead of the whole field.

---

> ### Author Response · Authors · 2024-12-02
> **Reply to Reviewer Kk5d**
>
> We appreciate the further reply. However, we want to clarify that we have replied to many of the concerns from reviewer Kk5d with additional explanations and citations of significant papers (like Nature Reviews Neuroscience, ICLR, and Neuron) to support our points and added figures and experiments. We hope that the reviewer can further consider the rating.
>
> **For biological motivation**, we aim to improve the inference of brain connectivity in disease progression models, which serves as the substrate for neurodegenerative disease propagation in human brains according to various connections among brain regions. This addresses a major need in computational medicine. Recent reviews, for example (Young et al. Nature Reviews Neuroscience 2024 [1]; Vogel et al. Nature Reviews Neuroscience 2023 [2]), specifically highlight the lack of accuracy in connectomes as a key issue with mechanistic spreading models for AD and other neurodegenerative diseases.
>
> **For the clinical benefit:**
> Disease progression models provide staging systems to position individual patients along a cohort-level trajectory. We added Figure 9 in Appendix A.5 to illustrate how the cohort-level long-term trajectory is constructed and used to determine a new subject's disease stage relative to the cohort, which is critical for diagnosis. This stage reflects pathology progression, informing treatment decisions and care planning. Additionally, integrating graph inference with LLM enhances the graph as a substrate for mechanistic disease progression, offering deeper insights into Alzheimer's disease (AD) mechanisms. This includes understanding interactions between brain regions, as shown in the inferred graph with LLM reasoning. Such insights help identify key drivers of tau spreading between brain regions, enabling more effective treatment strategies.
>
> **For the metric to evaluate new "findings"**, in the downstream task of tau modelling using the inferred graph, we used the alignments (Sum Squared errors and person R correlation) of the tau observation with the model prediction on the "new arrival" subjects from the test subjects as the metrics. In Figure 2 and Table 1, we have demonstrated that the LLM-graph can remain a good prediction of tau on test subjects with less number of edges in the graph compared with the usage of the biological graph, such as the single structural connectome or the combination of several biological graphs like the structural, functional and geodesic brain connectivities, that people used before in mechanistic modelling for AD progression, which shows that the LLM-graph added in values compared with the existing graph by remaining most significant edges. The improved convergence, stability, and identifiability in the data-driven neural graphical modelling (Table 2, Appendix A.2) also indirectly prove that the constraints of LLM bring about advances.
>
> Regarding **the concern about the meaningful biological knowledge from LLM for AD**, Abdulaal et al. ICLR 2023. demonstrate that LLMs can effectively complement traditional data analysis approaches through their Causal Modelling Agent (CMA). They successfully identified biologically plausible causal relationships (validated through both statistical analysis and existing literature) by combining LLM-based reasoning with data-driven modelling. Their findings regarding TREM2 and sex-specific mechanisms to tau pathology for AD align with and help synthesize existing evidence from multiple studies.
>
> Finally, we want to emphasize that our work does not aim to use LLM to uncover all of the unknown mechanisms of AD but instead take a step to demonstrate how judicious use of LLMs can improve the basic components of disease progression models, specifically the estimated brain connectomes, and bring us closer to providing insight into disease mechanisms via computational modelling. In our prompt, we carefully hint at LLM to think about the synergistic effect of existing biological graphs as a way to constrain LLM. We show in Appendix A.3 that the LLM graph shares multiple patterns from different biological graphs, thus validating the biological plausibly. We think it's valuable to explore the potential of LLM in this field step by step with careful experiment designs.
>
> Thus, we would appreciate it if the reviewer could reconsider the rating for our work.
>
> **References**
>
> [1] Young, A. L. et al (2024). Data-driven modelling of neurodegenerative disease progression: thinking outside the black box. Nature Reviews Neuroscience, 25(2), 111-130.
>
> [2] Vogel, J. W. et al (2023). Connectome-based modelling of neurodegenerative diseases: towards precision medicine and mechanistic insight. Nature Reviews Neuroscience, 24(10), 620-639.
>
> [3] Abdulaal, A. et al (2023). Causal Modelling Agents: Causal Graph Discovery through Synergising Metadata-and Data-driven Reasoning. In The Twelfth International Conference on Learning Representations.

---

> > ### Comment · Reviewer_Kk5d · 2024-12-02
> >
> > I am still concerned about the scientific rigor of this approach (but I am open to other reviewers). One major reason to study brain connectome is to have a holistic view of whole brain. Using LLM to query single connectivity is questionable. Furthermore, how the propagation mechanism is an open question. Currently, there is no in-vivo imaging technique that allows us to measure the region-to-region pathway as a model of the structural/functional connectomes.

---

> ### Author Response · Authors · 2024-12-02
> **Further response**
>
> We thank the reviewer for the timely feedback and valuable concerns.
>
> We would like to clarify that **in vivo imaging techniques do exist that enable the measurement of region-to-region pathways**, forming the basis for structural and functional connectome models. For instance, Diffusion MRI (dMRI) provides the ability to map white matter tracts, a cornerstone for understanding structural connectomes. Advanced tractography methods, such as constrained spherical deconvolution and probabilistic tractography, facilitate region-to-region pathway analysis, although they come with known limitations like resolving crossing fibres. Similarly, resting-state fMRI (rs-fMRI) captures correlated activity between brain regions, enabling the modelling of functional connectomes. Analytical tools, such as graph theory, are widely used to extract insights from these functional and structural networks. Moreover, multi-modal approaches like Structural-Functional Coupling (SFC) and PET-MRI fusion improve the resolution and integration of structural and functional data, advancing the potential to study region-to-region connections.
>
> By integrating knowledge from structural, functional, and other multimodal graph knowledge with the mechanisms of disease progression from sufficient literature and other materials, we posit that the LLM has the potential to exceed the limitations of single-modality biological graphs and provide a holistic perspective.
>
> **Regarding the propagation mechanism**, we acknowledge that this remains a challenging question in the field. To address this, our approach incorporates two complementary methodologies. First, we use mechanistic models, which explicitly describe disease progression with pre-defined mechanisms. Second, we demonstrate using a neural graphical model (described in Sections 2.5.2 and 3.3), which removes all the mechanistic assumptions and instead leverages a black-box neural network to learn progression trajectories and infer the graph structure (showing whether the dependence of one region to another exists) simultaneously. Our findings demonstrate that incorporating the LLM as a constraint improves robustness, interpretability, and convergence stability compared to purely data-driven approaches, which often yield inconsistent graph outcomes, as shown in Figure 3.
>
> Thus it would be highly appreciated if you could further consider the rating.

---

> > ### Comment · Reviewer_Kk5d · 2024-12-02
> >
> > Sorry, I was not clear. I was referring to the in-vivo pathways of pathology spreading.

---

> ### Author Response · Authors · 2024-12-02
>
> Thank you for clarifying your comments. We want to emphasize that connectome-based methods are widely recognized as mainstream approaches for modelling tau propagation, as highlighted in several recent Nature Reviews Neuroscience articles listed before. These methods effectively leverage biological connectomes, such as structural connectomes, to understand and simulate the spread of pathology in Alzheimer's disease. Evidence from optical imaging in post-mortem human brains also supports the hypothesis of trans-synaptic tau spread (Spires-Jones et al., Neuron, 2023).
>
> On the other side, we also demonstrate using a neural graphical model. This approach **removes all mechanistic assumptions** and employs a black-box neural network to learn progression trajectories and simultaneously infer the regional interactions using a graph representation. This includes identifying whether dependencies exist between different brain regions and providing a complementary perspective to traditional mechanistic models.
>
> Tau-PET imaging is an in-vivo technique for capturing snapshots of tau accumulation at specific time points. By aligning these snapshots along a temporal axis using our proposed framework, we can model the dynamic progression of pathology. Importantly, we demonstrate that leveraging LLM expertise enhances the alignment of dynamical Tau modelling compared with real PET measurements, similar to other mainstream papers exploring Tau pathology, thereby serving as a downstream validation of the inferred graph structure.
>
> Please reconsider the recognition of our contribution.
>
> **Additional References**
> Spires-Jones, T. L. (2024, July). Synaptic oligomeric tau in Alzheimer’s disease—A potential culprit in the spread of tau pathology through the brain. In Alzheimer's Association International Conference. ALZ.

---

### Meta-Review · Area_Chair_GiRN · 2024-12-20

**Metareview:**

The reviewers raise significant concerns about the scientific validity and clinical applicability of the proposed LLM-guided framework for modeling Alzheimer's disease progression. Key issues include the lack of rigorous evidence supporting the biological relevance of LLMs, small datasets, insufficient experiments, limited baseline comparisons, and the inability to address disease heterogeneity. While the rebuttal provided some clarifications, it failed to resolve doubts about the reliability of the LLM-inferred graphs and their clinical feasibility.

**Additional Comments On Reviewer Discussion:**

The reviewers have pointed out some significant drawbacks fundamental to the clinical feasibility of the LLM on AD understanding. These concerns by some reviewers (Kk5d, erUT) were not resolved by the author responses as the reviewers strongly believe that the work exhibits fundamental issues. Reviewer YR4J has raised the score but with some doubts left.

---

### Decision · Program_Chairs · 2025-01-22

Reject